# High capacity in G protein-coupled receptor signaling

Amiran Keshelava[1], Gonzalo P. Solis[1], Micha Hersch[2,3], Alexey Koval[1], Mikhail Kryuchkov[1], Sven Bergmann [2,3,4] & Vladimir L. Katanaev[1,5]

G protein-coupled receptors (GPCRs) constitute a large family of receptors that activate intracellular signaling pathways upon detecting specific extracellular ligands. While many aspects of GPCR signaling have been uncovered through decades of studies, some fundamental properties, like its channel capacity—a measure of how much information a given transmission system can reliably transduce—are still debated. Previous studies concluded that GPCRs in individual cells could transmit around one bit of information about the concentration of the ligands, allowing only for a reliable on or off response. Using muscarinic receptor-induced calcium response measured in individual cells upon repeated stimulation, we show that GPCR signaling systems possess a significantly higher capacity. We estimate the channel capacity of this system to be above two, implying that at least four concentration levels of the agonist can be distinguished reliably. These findings shed light on the basic principles of GPCR signaling.

[1] Department of Pharmacology and Toxicology, Faculty of Biology and Medicine, University of Lausanne, 1011 Lausanne Switzerland. [2] Department of Computational Biology, Faculty of Biology and Medicine, University of Lausanne, 1011 Lausanne Switzerland. [3] Swiss Institute of Bioinformatics, 1015 Lausanne Switzerland. [4] Department of Integrative Biomedical Sciences, University of Cape Town, Cape Town 7925, South Africa. [5] School of Biomedicine, Far Eastern Federal University, Vladivostok 690922, Russian Federation. Amiran Keshelava, Gonzalo P. Solis and Micha Hersch contributed equally to this work. Correspondence and requests for materials should be addressed to S.B. (email: sven.bergmann@unil.ch) or to V.L.K. (email: vladimir.katanaev@unil.ch)

G protein-coupled receptors (GPCRs) are the biggest receptor family in the animal kingdom, with about 1000 GPCRs encoded by the human and other mammalian genomes[1]. The structural versatility of GPCRs has allowed this type of transmembrane protein to evolve into efficient transducers of a variety of signals (e.g., light, small molecules, and lipoglycoproteins) across the plasma membrane. More than half of all marketed drugs target GPCRs or their signaling pathways[2]. Thus, better understanding of GPCR signaling is crucial for biology and medicine.

On the intracellular side, heterotrimeric G proteins are the main and immediate transducers of activated GPCRs[3]. In resting state, these proteins exist as complexes of α, β, and γ subunits, where the α-subunit is bound to GDP. The $G\alpha^{GDP}\beta\gamma$ complex can associate with GPCRs. Upon ligand interaction, GPCRs elicit the exchange of the guanine nucleotide on the heterotrimeric G protein, such that GDP bound to Gα prior to activation is substituted with GTP. This exchange triggers dissociation of $G\alpha^{GTP}$ from the βγ heterodimer; both components are then capable of transmitting the signal to downstream effector proteins. The intrinsic GTPase activity of Gα eventually leads to hydrolysis of GTP to GDP; this activity is further accelerated by GAPs (GTPase-activating proteins). Certain effectors of $G\alpha^{GTP}$ (e.g., phospholipase Cβ (PLCβ)) can mediate the GAP activity; alternatively, this activity is provided by dedicated RGS (regulator of G protein signaling) proteins[4]. Deactivated $G\alpha^{GDP}$ can interact with βγ to recreate the heterotrimetic G protein, which can again be activated by the GPCR. Alternatively, $G\alpha^{GDP}$ can reload with GTP (an activity, which is also suppressed by some RGS proteins) and continue signaling[5].

Despite several decades of GPCR signaling studies, some basic aspects of signal transduction by this type of receptors remain insufficiently characterized, in particular those pertaining to robustness, redundancy, specificity, signal amplification, and noise suppression. Some of these aspects can be addressed through information theory, which was originally proposed by Claude E. Shannon in 1948 to find fundamental limits on signal processing and communication operations[6–8]. A fundamental concept of information theory, channel capacity is the property of an information-transmitting system, characterizing the maximal amount of information that this system can reliably transmit in a given time. The higher the channel capacity, the more information it can transmit. Channel capacity is measured in bits. A channel capacity of one bit describes a system capable of reliably transmitting a simple on or off signal. A channel capacity of $n$ bits reliably resolves $2^n$ values.

During recent years, the use of information theory has led to important insights regarding noise control in intracellular signal transduction[9–11]. Specifically, Levchenko and co-workers[9] assessed the channel capacity of intracellular signaling pathways, such as TNF-NFκB signaling in mouse fibroblasts. They estimated that this capacity in an individual cell is close to 1 bit (0.92 bits). Using published data from other laboratories[12], the authors calculated that in other signaling pathways the channel capacity of individual cells is also close to 1 bit (e.g., 1.22 for the $Ca^{2+}$ response of RAW264.7 macrophages to uridine diphosphate stimulation through the P2Y-type GPCRs[9]). This low channel capacity implies that an individual cell can reliably transmit only a "yes-or-no" information regarding the received signal, but is unable to receive any information beyond this about the concentration of the signal. Subsequent works by other teams have arrived at a similar conclusion about the low capacity of intracellular signaling systems[12,13]. The known ability of cells to "read" different concentrations of the signal is held by the authors to stem from the collective cell behavior, where cell ensembles have a higher channel capacity (e.g., 14 cells can yield up to 1.8 bits of information)[9].

These studies[9,12] lay important ground for the application of information theory to intracellular signaling; yet, the experimental approach used to obtain the data and estimate the channel capacity has certain limitations. Specifically, in these experiments measuring the response strengths of several hundreds of cells responding to different concentrations of the ligand, any individual cell was never exposed to more than a single concentration of the ligand. Analysis of the corresponding data revealed that no statistically significant difference between any two tested concentrations of the ligand, except for "no" ligand and "some" ligand, could be identified. This finding, however, may well be confounded by the fact that different cells respond with different strength to the same concentration of the stimulus. Thus, measurements averaging over many cells can effectively result in an estimate of the channel capacity that is much smaller than the intrinsic capacity of a single cell[14,15]. However, it has also been demonstrated that the strength of the cell response for a given stimulus concentration, be it big or small, is conserved in repeated stimulations (with an average correlation of about 0.9), meaning that the system is quite robust and has a low level of intrinsic noise[14]. Interestingly, extending the dimensionality of the readout by recording single-cell dynamic responses to a single stimulation led to an estimated maximal mutual information between the ligand concentration and the (multidimensional) dynamic response to well above 1 bit[15]. Nevertheless, despite this mounting evidence for larger than one channel capacities, direct quantification of the channel capacity of intracellular signaling pathways using a range of stimulations is still lacking.

Here we design an experimental approach addressing this issue by recording single-cell responses of a GPCR signaling pathway to multiple stimulations with multiple concentrations, allowing the direct estimation of its channel capacity for each individual cell along with the reproducibility of its response. We find that the GPCR signaling has a channel capacity that is considerably higher than previously estimated from unidimensional (and even multidimensional) single-cell or multiple-cell recordings. This high-capacity signaling has important implications to the physiology of GPCR-mediated cellular activities.

## Results

**Collective cell response.** Averaged cell responses to an external stimulus, which may be considered as a collective cell behavior[9], are well known to cover a certain concentration range (called dynamic range) where cells reliably respond differently to different ligand concentrations (see e.g. ref. [14], also see Fig. 1a, b). We started by reproducing this feature in our experimental setup. From our data, it is apparent that the channel capacity of the response of relatively small populations of 20–30 cells is far greater than one: not only do the cells determine that they have been activated by some ligand concentration, but the strength of the response varies as a function of the ligand concentration (Fig. 1a, b). This raises the question, how do individual cells respond to increasing ligand concentrations. Two alternative possibilities exist, both explaining the depicted population cell response (Fig. 1b–d). The first implies that the cells have different thresholds of activation, each leading to a "yes-or-no"-like response. In this scenario, the channel capacity of individual cells (Fig. 1c) is close to one. This situation is expected based on previous studies[9,12,15]. In contrast, if each cell possesses a range of ligand concentrations where the responses to these concentrations are reliably different (dynamic range), as shown in Fig. 1d, even though this range is individual for each cell, then the channel capacity of the individual cell is significantly greater than one.

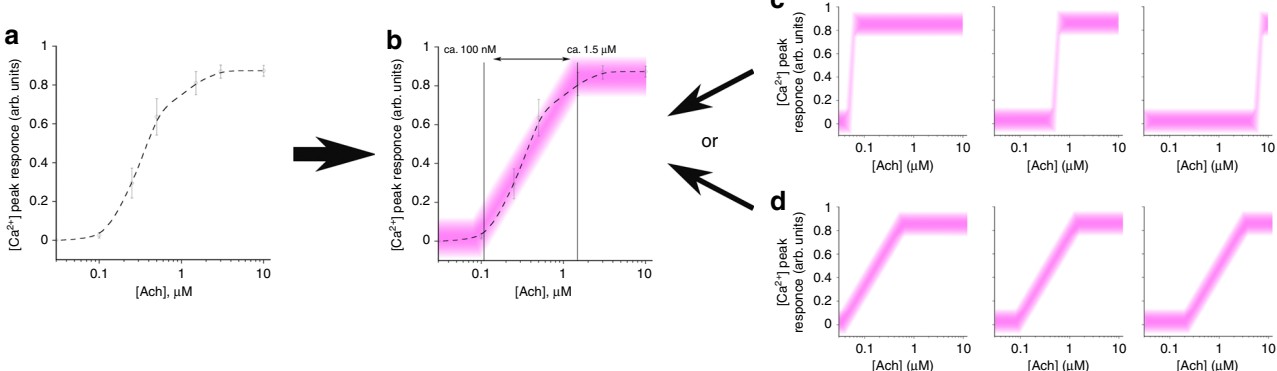

**Fig. 1** Population and individual cell responses. **a** Population of HEK293 cells elicits an intracellular Ca$^{2+}$ response to increasing concentrations of acetylcholine (Ach). Data are provided from cell responses averaged over four independent experiments, randomly selected from the full data set, 20–30 cells in each experiment, as mean ± sem. **b** This response has a certain dynamic range (marked by the vertical lines and double arrow), where the response strength is proportional to Ach concentration. **c**, **d** Two scenarios behind this population response can be envisioned. In the first **c**, each cell may possess a sharp threshold of a "yes-or-no" response, shifted differently along the agonist concentration scale. Alternatively **d**, each cell has its own dynamic range, also shifted differently along the agonist concentration scale. Broadness of the pink lines schematically illustrates the noise in the individual cell responses, provided by deviations from the mean response strength in the experiment whereby cells are stimulated multiple times

**Single-cell response to repeated stimuli**. To establish a suitable system for quantifying the channel capacity in GPCR signaling in individual cells, we selected HEK293 cells—a workhorse of cell biology, whose GPCR, heterotrimeric G protein, and GAP transcriptome is well characterized[16], permitting the selection of the proper GPCR pathway to study. We focused on the muscarinic acetylcholine receptor M3R—the only member of the acetylcholine (Ach) GPCR group expressed in HEK293 cells[16]. In vivo, M3R is expressed in a number of tissues[17] and regulates many important physiological and pathological conditions, including smooth muscle contraction and dilation of peripheral blood vessels[18], cardiac function and heart disease[19], whole-body metabolic activities[20], insulin secretion[21], bone formation[22], tumor formation in the gastrointestinal tract[23], T-cell dysfunctioning in Sjögren's syndrome[24], and others[25]. The crystal structure of M3R bound in a complex with its antagonist tiotropium (clinically used for bronchodilation and against chronic obstructive pulmonary disease[26]) paves the way for rational design of drugs targeting this GPCR[27].

M3R activates heterotrimeric G proteins of the G$_q$ family. GTP-loaded Gαq subunit in turn activates PLCβ, two isoforms of which (PLCβ1 and PLCβ3) are expressed in HEK293 cells[16]. PLCβ cleaves phosphatidylinositol 4,5-bisphosphate (PIP$_2$) into diacylglycerol and inositol 1,4,5-trisphosphate (IP$_3$), the latter being responsible for the opening of calcium stores in the endoplasmic reticulum and the rise in intracellular [Ca$^{2+}$][28].

We used the dye Fura-2 AM to monitor intracellular [Ca$^{2+}$] in individual live cells in a setup permitting multiple pulses of the activator (Ach) at different concentrations. In this setup, a field containing 10–35 cells is microscopically captured, and intracellular [Ca$^{2+}$] is monitored by dual-wavelength fluorescence recording (see Methods section, Fig. 2a, and Supplementary Movie 1). Prior to performing the experiments permitting channel capacity estimation through repeated stimulations with different Ach concentrations, we analyzed the response of individual cells to repeated activations (at least 20 times) with a single selected agonist concentration of 250 nM. This initial experiment led us to draw the following important conclusions. First, we find that different cells respond with different strengths to the same stimulus concentration (Fig. 2a, b).

Second, it is also apparent that these different cellular responses are reproducible over at least 20 stimulations (Fig. 2b).

As a more formal measure of this reproducibility, we computed the correlation across stimulations of individual cell responses over >20 cells for at least 20 stimulations, where the strength of each single response $n$ was correlated with that of the next one, $n + 1$. This analysis shows (Fig. 2c) that the response strength, being different among individual cells, is highly reproducible within cells with a correlation $r = 0.999$. This value is higher than the $r = 0.9$ estimate obtained upon pairwise single-cell activation in another GPCR signaling system[14]. This high reproducibility also indicates that any noise originating from the experimental imprecision (such as pipetting or fluorescent recording errors) is minimal in the experimental setup we constructed.

Third, this analysis also reveals that the slope of the regression line of the $(n + 1)$th to the $n$th response is significantly smaller than 1 (0.99, $p$ value $< 6 \times 10^{-8}$, linear regression test, Fig. 2c), which indicates that the response tends to decrease with each stimulation. Indeed, over the series of stimulations, the [Ca$^{2+}$] peak response height becomes progressively smaller (Fig. 2b, d). Linear regression analysis in log space indicates that this decrease is modest but highly significant, slightly below 1% (with a 95% confidence interval of (0.89% 1.06%)) for each stimulation pulse, or about 18% drop in response strength over 20 rounds of stimulation (Supplementary Data 1–4). This phenomenon is expected and is likely related to the issue of desensitization (adaptation) well-known for the GPCR-mediated signaling pathways[29], as well as to intracellular signaling in general. At higher agonist concentrations, this phenomenon is also noticeable with less stimulations (see below, Supplementary Figure 1a).

The 250 nM concentration leads typically to low/intermediate cellular [Ca$^{2+}$] response (Figs. 2a–c and 3a). In contrast, we find that μM concentrations already give a saturating response, as subsequent stimulations with ionomycin, which releases calcium ions from intracellular stores and thus displays the maximal possible cell response to stimulation[30], gives a comparable peak height with a broader width, as expected[30] (Fig. 3a, b).

All these preliminary experiments served as the basis for designing our experimental setup for the channel capacity estimation. In this setup, cells are repeatedly (five times) stimulated in the following increasing sequence of Ach concentration: 100 nM, 250 nM, 500 nM, 750 nM, 1.5 μM, 3 μM, and 10 μM. This empirically selected set of concentrations allows detecting a minimal observable cell response at lower

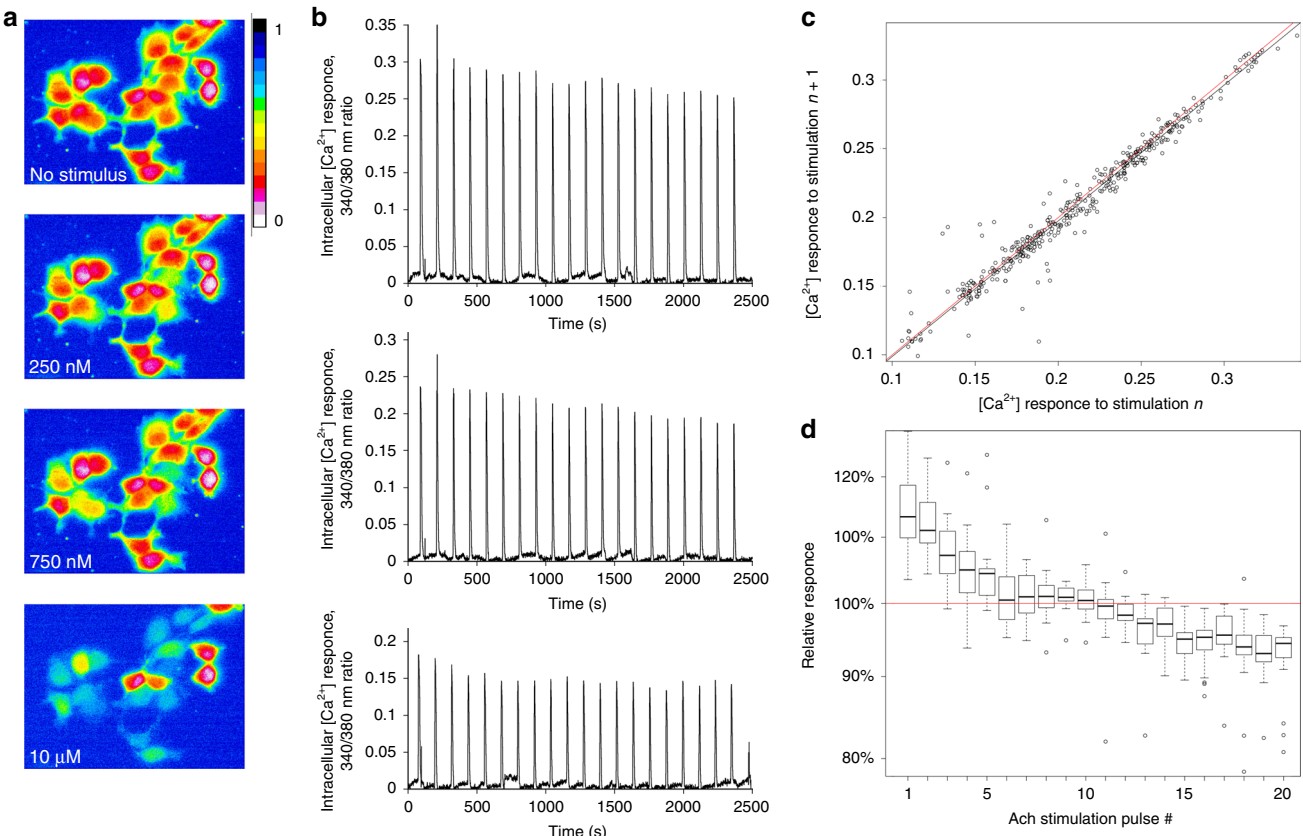

**Fig. 2** Single-cell analysis of response to repeated stimulations. **a** Example of the experimental setup. A field of HEK293 cells loaded with Fura-2 AM is microscopically captured and imaged after buffer only perfusion (no stimulus), or perfusion with 250 nM, 750 nM, and 10 μM Ach. The intracellular $Ca^{2+}$ response is given as the 340/380 nm ratio in the pseudocolor (scale given to the right of the top panel). Cell stimulation is seen as the change in the cell color from red (ratio close to zero) to yellow or blue. The images given here are captured from Supplementary Movie 1. **b** Three examples of individual cell's responses to repeated (×20) stimulations with 250 nM Ach. Well-responding cells are selected for clarity. It is apparent that: (i) different cells respond with different strengths to the same agonist concentration; (ii) the response strength of the same cell is reproducible over multiple stimulations; (iii) a slight trend of a decrease in the response strength is seen from early pulses to later pulses of stimulation. **c** Correlation analysis of the reproducibility in individual cell's responses to repeated stimulations. Each circle represents a pair of consecutive response to the same stimulation in the same cell. The responses are highly conserved but tends to dampen as can be observed from the linear regression (black line), which has a slope significantly lower than one (red line shows the diagonal). **d** Cells adapt (desensitize) their $Ca^{2+}$ response over time, as repeated equal stimulations lead to a slight but significant dampening of the response

concentrations and to plateau at the maximal cell response for the highest concentrations. Ten-second-long stimulations with each Ach concentration were repeated five times with a 110-s gap in between when the cells were rinsed with a buffer. This temporal protocol was selected taking into account the cell Ach response kinetics: with our experimental setup, it takes 7–9 s for a cell to reach the peak value of the calcium response (Figs. 2b and 3a, b).

The overall duration of each experiment is thus $(10 + 110\,\text{s}) \times 5 \times 7 = 70$ min. This time is below the time required for the transcriptional and translational response to an external stimulus, including that mediated by GPCRs, to be manifest (which takes about 2 h for the first appearance of the response, and 3–6 h to reach the response maximum)[31]. Thus, any feedback changes in expression of M3R or components of its signaling pathway will not happen during the run of the experiment.

Stimulation with increasing Ach concentrations, rather than randomized agonist concentrations, is essential to minimize both the impact of desensitization (Fig. 2b–d and Supplementary Figure 1a), and the risk of interference of any traces of Ach left in the system from the previous pulse with the current stimulation pulse.

**High channel capacity of the individual cell response**. The selected experiment setup was applied to 433 cells (10–35 cells in each of 27 independent experiments), of which examples are provided in Figs. 2a and 3a, and Supplementary Movie 1 (the full set of data is available as Supplementary Data 1). As can be seen in Fig. 3a, while single cells typically do not respond to the lowest Ach concentrations, they reproducibly respond roughly in proportion to the Ach concentration for the higher concentrations, and typically reach a plateau for Ach concentrations greater than 1.5 μM. It is evident from these examples that a cell can reproducibly transmit information, from the extracellular milieu to its cytoplasmic $Ca^{2+}$ mobilization, about how much stimulus it has received. Further, comparison of many individual cell responses confirms our expectation that, while each individual cell possesses its own dynamic range of ligand concentrations, these ranges can be "shifted" toward lower or higher concentrations in different cells (Fig. 3c), very similar to how we have schematically depicted this scenario in Fig. 1d.

Formal analysis of this large panel of data, collected from numerous individual cells (Methods), confirms that the Ach M3R GPCR response has a channel capacity significantly exceeding the previous estimations[9,12,15]. Indeed, for the $n = 433$ cells that were

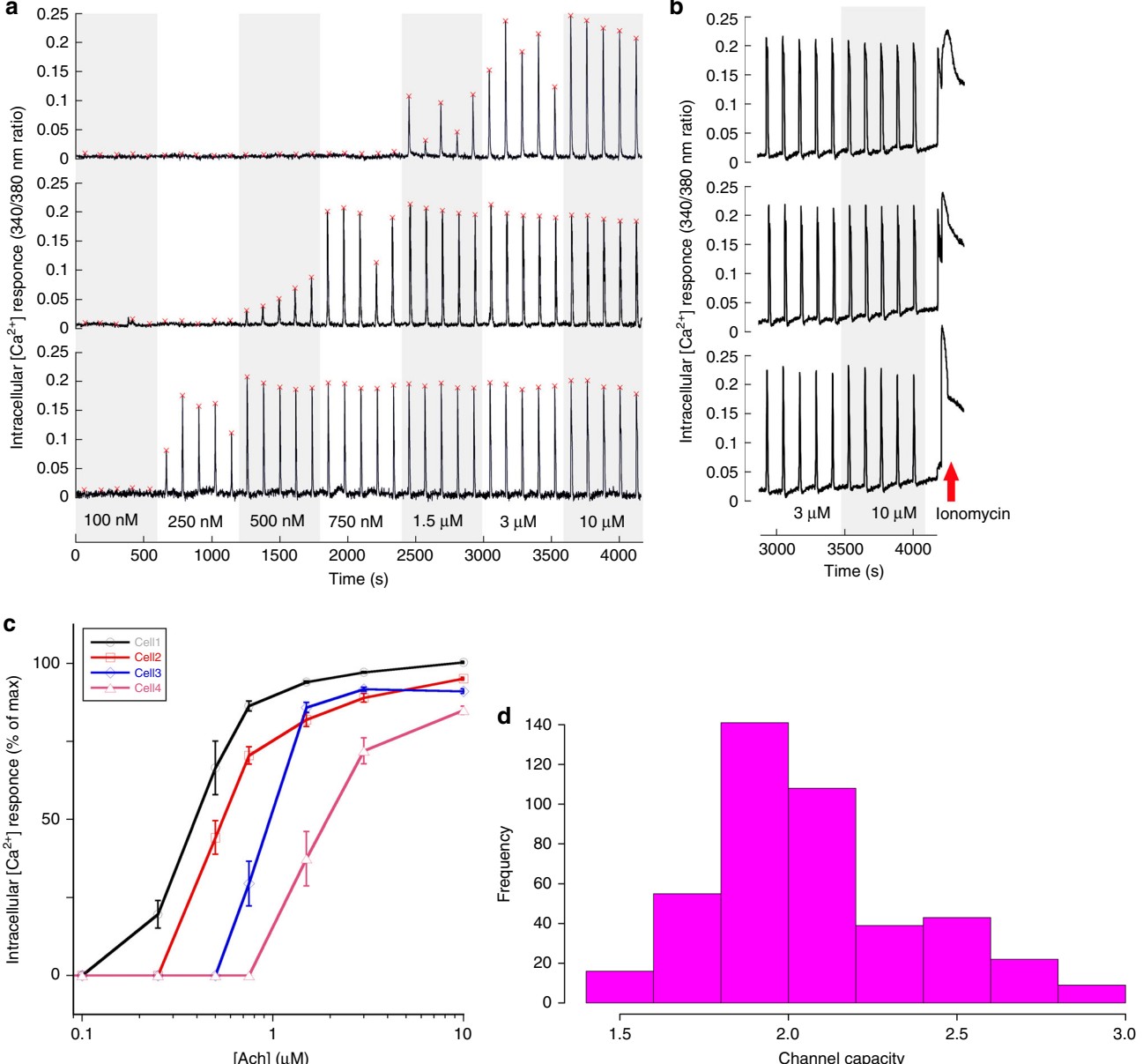

**Fig. 3** Individual cells have individual dynamic ranges and high channel capacity. **a** Three examples of different single cell's Ca²⁺ responses to repeated stimulations with increasing concentrations of Ach. It is evident that some cells respond only to high Ach concentrations (top example), while others are sensitive already to the low concentrations but rapidly reach the plateau in their response strength (bottom example). The example in the middle is an intermediate. See also Fig. 2a. **b** To prove that cells indeed reach the plateau in their response capacity at high Ach concentrations, the Ca²⁺ ionophore ionomycin (2 µM) was added at the end of the experimental series in some experiments. Three examples are provided, illustrating the typical cell response to ionomycin. **c** Examples of cells possessing dynamic ranges shifted along the agonist concentration scale but all capable of responding reliably and differently to different agonist concentrations. **d** Histogram of estimated channel capacities estimates. The average estimate of channel capacity is $2.06 \pm$ 0.31 (mean ± sd) bits, whereas the lower bound lies at $1.65 \pm 0.18$ (mean ± sd; see Supplementary Figure 1b)

analyzed, and using a parametric distribution approach, we find an average estimate of the channel capacity at $2.06 \pm 0.31$ bits (mean ± sd, see Fig. 3d for the distribution). Without using parameter interpolation, we find a lower bound estimate of $1.65 \pm 0.18$ bits (see Supplementary Figure 1b, and Methods for additional details).

This conclusion highlights that the GPCR signaling pathway we analyzed here can reliably transmit information about different concentrations of agonist. Furthermore, given the fact that the dynamic range of individual cells is shifted left or right within the selected concentration range (Fig. 3c), our

experimental setup somewhat underestimates the channel capacity for many cells. Accordingly, when channel capacity values of individual cells were plotted against the $EC_{50}$ of their response, the polynomial fit of this distribution clearly shows that higher estimates of the channel capacity are calculated for the cells with $EC_{50}$ in the middle of the range of ligand concentrations (Supplementary Figure 1c). The overall magnitude of this effect is rather small, reaching only 10–20% of mean channel capacity value. In other words, the M3R signaling system can reliably transmit information of at least four different concentrations of the stimulus (including "no stimulus").

## Discussion

Analyzing GPCR signaling in individual cells, we came to the conclusion that the channel capacity of this signaling system is above 2. Dynamics of intracellular signal transduction and integration of its outcomes over time may further increase this channel capacity estimate[9,15,32]. Our results differ from previously published estimates because we probed the system at the single-cell level whereby single cells were probed multiple times with different agonist concentrations, as opposed to the previous approaches to the problem, which involved single stimulations of cells[9,12,13,15]. Interestingly, our lower bound results are in line with the channel capacity of around 1.5 estimated from the single-cell dynamic response published in the study by Selimkhanov et al.[15], further validating its point that (given enough cells) the dynamic response partly compensates (in terms of mutual information) for the ignorance of the actual cell state in the estimation of the channel capacity. In other words, measuring single-cell responses to single stimuli at multiple time points indeed helps to some extent distinguish between extrinsic (i.e., across cells) and intrinsic (i.e., within cells) response variability.

In a cell living in a complex environment and responding to a multitude of external and internal factors (e.g., cell–cell interactions, growth factors, cell growth and division cycle, and so on), each of the steps of the signaling pathway (channel) we studied may be influenced by these other factors in the form of expression level, post-translational modifications, and localization of the protein components, as well as production/release and degradation/removal of the second messenger components. Given the limited experimental control over these factors, they will be confounding for the channel capacity assessment of the signaling pathway and increase the noise in the measurements. This variability further affects the concentration range to which a particular cell responds, "shifting" this range for some cells away from the range fixed in the experiment. These considerations further indicate that the high channel capacity we have calculated still underestimates the real capacity of intracellular signaling pathways.

The Ach-M3R-Gq-PLCβ-IP$_3$-Ca$^{2+}$ signaling pathway studied here in single HEK293 continues at levels below intracellular Ca$^{2+}$ to regulate multiple cellular activities, such as the activation of cellular kinases (e.g., different PKC isoforms) and their targets, cytoskeletal rearrangements, or transcription[33]. We do not know whether bifurcation of the signal-transmitting channel downstream of intracellular Ca$^{2+}$ splits the high channel capacity into lower capacity subchannels or whether the signaling pathway maintains the high capacity all the way down. Further, several of the intermediate components of the Ach-M3R-Gq-PLCβ-IP$_3$-Ca$^{2+}$ signaling system may have other effectors than those studied as the main signaling "highway" in our work. As examples, the M3R GPCR can activate β-arrestins in addition of the heterotrimeric Gq protein[25]; additional effectors of Gαq-GTP and Gβγ released from the Gq heterotrimer exist in addition to PLCβ[34]; and the second messenger IP$_3$ possesses other signaling outcomes in addition to opening intracellular Ca$^{2+}$ stores[28]. It is thus clear that a signaling network, instead of a single isolated pathway, exists in cells and can be compared to a network of roads of different importance (capacity): highways and regional roads exiting from and entering to these highways at different points[35]. In this analogy, the Ach-M3R-Gq-PLCβ-IP$_3$-Ca$^{2+}$ pathway we studied would represent a highway, whose channel capacity is measured as high on the selected long distance. Channel capacity measurements of the subsequent parts of this road map and of the in- and out-coming "regional routes" should be a matter of subsequent studies, which would require the establishment of the experimental and theoretical framework permitting the application of information theory to a network of intracellular signal transduction.

Our findings show that a single cell is endowed with high-capacity information transmission channels, permitting it to robustly respond to different levels of external stimulations. Individual dynamic ranges of different cells in a population are adapted to different absolute agonist levels. These conclusions provide the explanatory basis for the well-established processes in cell and developmental biology, such as tissue morphogenesis, whereby cells acquire different fates depending on the concentration of the morphogen received[36]. GPCR signaling equipped with high channel capacity must also be employed in cases such as chemotaxis of leukocytes, where cells need to possess the capacity of reliably reading different chemoattractant concentrations in order to move along the concentration gradient[37,38]. In contrast, certain GPCR signaling systems, which need to provide only a yes-or-no response, such as those controlling cell differentiation[39], may well possess a reduced channel capacity, similar to that estimated in the previous works[9,12,13,15]. It will be interesting to compare, using the single-cell approach introduced in our study, the channel capacity in these different signaling systems. Understanding of the underlying molecular mechanisms of (potentially) different channel capacities in these systems will provide further advances in the basic understanding of intracellular signal transduction. Furthermore, it will open the possibilities to synthetically construct signaling pathways with reduced or increased channel capacity.

Finally, our work shows that like many other systems, the GPCR system is adaptive, as the cell response is influenced by previous stimuli. As such, formally it is not the memoryless channel assumed by the channel capacity concept. As a first approximation, it is reasonable to discard the effect of adaptation, as has been done previously in the field. Indeed, correcting for adaptation by removing the median residual for each pulse number (Supplementary Figure 1a) only increases the channel capacity by 6% (Supplementary Data 4). However, it may be insightful to move beyond the concept of channel capacity and consider the amount of information that can be transmitted by such adaptive systems, which is likely to further increase the channel capacity. This would open interesting questions into adaptive information transmission in biological systems[40].

In summary, we provide insights into the basic principles of GPCR signaling. We expect that our investigation will inspire further work on the molecular mechanisms of the signaling systems with implications for healthy and pathological biology of the cell.

## Methods

**Cells and calcium recording**. HEK293 cells were freshly obtained from ATCC (catalog #CRL-1573; Middlesex, UK) for the sake of reproducibility with the GPCR, G protein, and GAP expression data of previous studies[16]. Cells were grown in DMEM (Thermo Fisher Scientific, Waltham, MA) supplemented with 10% FCS and penicillin–streptomycin at 37 °C and 5% CO$_2$. Trypsinized cells were seeded on poly-L-lysine-coated coverslips for 24 h and then loaded with 5 μM Fura-2 AM (Biotium, Fremont, CA) in DPBS without calcium or magnesium (Thermo Fisher Scientific) for 30 min, followed by washing in HBSS supplemented with 10 mM HEPES pH 7.4 (HBSS-HEPES; Thermo Fisher Scientific). Cells were mounted in a perfusion chamber (Warner Instruments, Hamden, CT) and recorded (excitation (100 ms): 340, 380 nm, emission 520 nm) at one image per s in a VisiScope Cell Explorer System (Visitron Systems, Puchheim, Germany) equipped with a Plan Neofluar ×40/1.3 oil objective on an Axio Observer.A1 microscope (Zeiss, Jena, Germany), a CoolSNAP HQ2 CCD camera (Photometrics, Tucson, AZ), a Visi-Chrome Polychromator with a Xenon-lamp 75Watts, and the MetaFluor Fluorescence Ratio Imaging software (Molecular Devices, Sunnyvale, CA). Solutions were changed with the cFlow flux controller (Cell MicroControls, Norfolk, VA) and the Solution Changer Manifold MSC-200 (Bio-Logic, Seyssinet-Pariset, France), and cells were stimulated with seven different acetylcholine (Ach; Sigma-Aldrich, St. Louis, MO) concentrations in HBSS-HEPES: 100 nM, 250 nM, 500 nM, 750 nM, 1.5 μM, 3 μM, and 10 μM. This set of concentrations was selected empirically. It allows detecting a minimal observable cell response at lower concentrations and to reach a plateau of the maximal cell response for the highest concentrations.

Ten-second-long stimulations with each Ach concentration were repeated five times with a 110-s gap where the cells were rinsed with HBSS-HEPES. This temporal protocol was selected taking into account the cell Ach response kinetics: with the used experimental setup it takes 7–9 s for a cell to reach the peak value of the calcium response. Ionomycin (2 μM; Sigma-Aldrich) was added at the end of some experiments.

The whole set of raw data for 433 cells obtained during this work is provided as Supplementary Data 1.

**Data analysis and software**. Data analysis was performed with a script developed in-house in MATLAB 2014a (MathWorks, Natick, MA) for Linux (with Signal Processing and Bioinformatics toolboxes); the script is provided as Supplementary Data 2.

The program employs the following algorithm:

1. Using an integrated MATLAB function, the program removes the background (which is rising during the course of the experiment, as can be seen in Supplementary Movie 1).
2. Program is searching for $x$ ($x = 35$ in this work) peaks (expected from $K$ ($K = 5$ in this work) repeated measurements from $N$ ($N = 7$ in this work) concentrations of ligand) on each trace produced by a cell (traces are cropped at the right to remove the post-stimulation zone).
3. Using the plotted traces, user selects the most representative trace with few or no artifacts and where most of the peaks are confidently defined. This trace is used to align peaks on the other traces and to extrapolate peaks identified in the left part of the graph where direct identification of the peaks is troublesome due to their proximity to background. This is achieved by searching the peaks at the mean distance between the most prominent peaks in the right part of the trace.
4. Any obvious recording artifacts identified as peaks on the trace are removed from further calculations (correspondingly, for some of the concentration groups $K$ will be <5).

For more detailed descriptions, one may refer to the comments in the script. The peak values, calculated using the script and used to estimate lower bounds on the channel capacity, are provided as Supplementary Data 3.

**Channel capacity calculations**. The channel capacity is the maximal rate of information that can be transmitted through a memoryless channel. This concept was developed within the framework of information theory initiated by Shannon[6–8].

More precisely the channel capacity $K$ is given by max I($r$, $c$), where $I$ is the mutual information between the channel input $c$ and the channel output $r$. In our setup, $c$ is given by the Ach concentration, while $r$ is the measured intracellular calcium response.

The mutual information between $c$ and $r$ has been shown to be equal to

$$I = \iint P(r,c)\log\left(\frac{P(r,c)}{P(r)P(c)}\right)\mathrm{d}r\mathrm{d}c, \tag{1}$$

where $P(r,c)$ is the joint distribution of $c$ and $r$.

Since $P(r,c) = P(r|c)P(c)$, we have

$$I = \iint P(r|c)P(c)\log\left(\frac{P(r|c)}{P(r)}\right)\mathrm{d}r\mathrm{d}c. \tag{2}$$

Moreover, by marginalizing over $c$, we have that $P(r) = \int P(r|c)P(c)\mathrm{d}c$, so that

$$I = \iint\left[P(r|c)P(c)\left(\log(P(r|c)) - \log\left(\int P(r|c)P(c)\mathrm{d}c\right)\right)\right]\mathrm{d}r\mathrm{d}c, \tag{3}$$

In sum, since the channel capacity is the max of $I$ over any input distribution $P(c)$, it is totally determined by $P(r|c)$, the distribution of the output conditioned on the input.

In the following, we estimate the capacity for each cell. To that end, for each cell i, we estimate $P_i(r|c)$ based on the data for each of the seven input concentrations $c_1$–$c_7$ that were tested. For the estimation of $P_i(r|c)$, we only have five points per cell and input concentrations (having more may bias the results because of adaptation). We therefore use the following model, for the output $r_{ijk}$ of cell i, stimulated with input $c_j$:

$$r_{ijk} = \mu_i(c_j) + e_{ik}(c_j), \tag{4}$$

where $e_{ik}(c_j)$ is a noise term, the distribution of which is shared across cells and specific to each input signal $c_j$, $\mu_i(c_j)$ is the cell specific mean response to the input signal $c_j$, and $k$ the replicate index.

The output concentration $r_{ijk}$ was defined as the log of the peak calcium concentration upon stimulation. Upon inspection, the distribution of the noise (in log space) appears to be symmetric but non-Gaussian and heavy-tailed (Supplementary Figure 2a, b). We therefore checked what heavy-tailed symmetric

distribution provides the best fit for the observed residuals computed as

$$e_{ik}(c_j) = r_{ijk} - \frac{1}{K}\sum_{k=1}^{K} r_{ijk}, \tag{5}$$

where $K$ is the number of replicates per input concentration per cell (usually five).

We found that the (scaled) $t$-distribution gives a good fit (Supplementary Figure 2a, b), and therefore used it in the model:

$$P(e_{ik}(c_j)) = \frac{\Gamma(0.5(\nu_j+1))}{\Gamma(0.5\nu_j)\sqrt{\pi\nu_j}\sigma_j}\left(1 + \frac{1}{\nu_j}\left(\frac{e_{ik}(c_j)}{\sigma_j}\right)^2\right)^{-0.5(\nu_j+1)}, \tag{6}$$

where $\Gamma$ is the standard Gamma function. The two parameters (scaling factor $\sigma_j$ and degrees of freedom $\nu_j$) of the model were then estimated using the maximum likelihood method[41], assuming all data are independent.

Finally, a standard optimization procedure was used to find the discrete distribution over the $c_j$ that maximizes $I$ (under the constraint that $\sum P(c_j) = 1$). In so doing, we only estimate a lower bound for the capacity, because when maximizing $I$ over the input distribution, we can only consider a subset of distributions, namely the ones that are zero for all $c \neq c_j$, ($j = 1, 2, ..., 7$). This means that we are maximizing over a subset of distributions, necessarily resulting in a lower bound estimate of the channel capacities. Indeed, with an experimental setup that probes only seven different concentrations, the highest estimate that is theoretically possible is $\log_2(7) \cong 2.8$ bits, even though a continuous noise-free channel would (theoretically) have an infinite capacity.

In order to alleviate this limitation, we performed a (piecewise linear) interpolation of parameters of the scaled student distribution in order to estimate $P_i(r|c)$ for concentrations that were not experimentally tested (Supplementary Figure 2c). We then found for each cell i the optimal $P_i(c)$ over the whole range of $c$. We thus have both a lower bound estimate that is limited by the amount of tested stimulus doses, and a more realistic estimate of the channel capacity based on parameter interpolation.

The R script implementing these analyses is provided as Supplementary Data 4.

**Data availability**. Data supporting the findings of this manuscript are available as Supplementary Data 1–4 and also can be obtained from the corresponding authors upon reasonable request.

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

## Acknowledgements

This work was supported by grants #31003A_138350 (Swiss National Science Foundation) and #KA2721/2-1 (Deutsche Forschungsgemeinschaft) to V.L.K. We thank Andrey Kajava, Georgy Karev, and Rico Rüedi for critically reading the manuscript, and Omar Alijevic for technical assistance.

## Author contributions

A.K. performed experiments and initial phase of mathematical analysis; G.P.S. established the experimental setup and contributed to experiments; M.H. analyzed the data and contributed to writing the manuscript; A.Ko. and M.K. contributed to data analysis; S.B. contributed to the study design and data interpretation; V.L.K. designed and planned the study, analyzed the data, and wrote the manuscript.

## Additional information

**Competing interests:** The authors declare no competing financial interests.

