## [Peer Review File · Nature Communications]

Reviewers' comments:

Reviewer #1 (Remarks to the Author):

Keshelava et al. present an evaluation of the information theory "channel capacity" concept in the context of a G-protein coupled receptor signaling system. Their experimental system is designed to observe single cells respond to multiple signaling events, in order to decouple responses from cell-to-cell variation. This is a very nice system that provides a new way to approach the question of channel capacity that has not been possible before, and the data obtained appear to be illuminating. The authors observe that individual cells can clearly respond differentially to gradations in stimuli, and they challenge previous claims of similar signaling systems being limited to binary discrimination of ligand presence. Overall, this is an elegant and technically sound experimental approach that is much more appropriate for making measurements of channel capacity than what has been done in previous work. However, it is somewhat disappointing that despite this excellent system, the authors have limited themselves largely to following the form of the previous analyses (Cheong et al. for instance) and reporting a slightly higher channel capacity (1.65 bits vs 1 bit), rather than taking the chance to recast this model more substantially. I know many people who feel that the Cheong et al "1 bit" study is unconvincing and poorly executed, even if the general concept of measuring information capacity is interesting one. It would be a shame if readers of this work took away only the conclusion that cells can distinguish 3 levels of stimulus rather than 2, when in fact it may be possible to go quite a bit further.

Major comments

1. It seems possible that a higher estimate of channel capacity could be made by altering the experimental setup. The authors note that the upper limit possible with 7 different doses is 2.8. However, for any given cell, most of the stimulant concentrations were saturating (either low or high), and it appears that few cells experienced more than 1 or 2 non-saturating doses, leaving their maximal observable capacity ~1.5-2. These limitations could be surpassed by tighter spacing of stimulant concentrations, spanning a suitable range for some cells to be sensitive. Theoretical extensions of the data could similarly offer estimates beyond the current limitations (see the following comment).
2. The estimate of capacity appears to be based on a continuous error model for each cell at each concentration. However, the final estimate is performed with only the number of discrete concentrations used in the experiment. This propagates the fundamental limitation of the study, that relatively few stimulant concentrations may be used. However, the question of theoretical channel capacity over a continuous range of concentrations may still be addressed approximately. The response models at discrete concentrations could be extended to (smoothly) estimate the response of each cell for any concentration, which may then be integrated to estimate the continuous channel capacity.
3. In discussing channel capacity, the definition of the "channel" for this study is only casually covered. The field stands to gain from careful consideration and discussion of how information theory concepts relate to real biological systems. In this case, the channel connects Ach outside of the cell to free Ca^{++} in the cytoplasm. However, Ach and the associated GPCR pathway are certainly not the only factors that control free Ca^{++} - other factors may control the responsiveness of Ca^{++} levels to Ach via modification of receptors, cofactors, signal transducers, and their expression levels. These complexities confound the channel capacity concept, which (as it is used here) is derived for single-input/single-output channels. Since we are unable to simultaneously measure all relevant inputs to free Ca^{++} , any inputs other than Ach would appear as noise, though in actuality they do not detract from real multi-input/multi-output channel. It would be helpful to discuss where the current experimental system stands in excluding the unmeasurable effects of these additional inputs.
4. It would also be helpful to consider and discuss other sources of variance, such as variance in each cell's actual exposure to Ach, pipetting errors, and noise in reporter measurements of Ca^{++} .

Minor comments

1. Editing for language and punctuation errors is needed.
2. Various graph axis labeling etc.: Ensure suitable font size, inconsistent Y-axis labeling in Fig. 2B, missing concentration label in Fig. 2A,
3. Formatting of the methods "Channel capacity calculations" section appears troubled, especially with regard to equations. E.g. an extra appearance of $P(r,c)$ at line 359, and a missing parenthesis at line 362

Reviewer #2 (Remarks to the Author):

The manuscript by Keshekava et al. addresses Ca signaling responses in response to stimulation of M3R receptors. The authors claim that the information transfer capacity of this signaling pathway is higher than previously reported, if one focuses on the single cell level dose response.

Unfortunately, in the opinion of this reviewer, the study, as presented, is misleading and the results and conclusions do not justify the messages presented. Below I present the reasons for this conclusion in no particular order.

1. The analysis treats the signaling system as essentially memories, thus assuming that the initial stimulus and response does not define the subsequent responses. However, the authors state themselves that the response is strongly adaptive, with each subsequent response decreasing in amplitude on the time scale of the experiment (for the same dose). In fact this is the reason the authors chose escalating doses in their analysis. This internal contradiction makes the analysis and the results unjustified.
2. The authors misrepresent the prior studies and do not place their result in the context of the actual biological response. In fact, biologically, the uncertainty of the selection of the cells that are involved in response is as important if not more important vs. the uncertainty of the dose response of an individual cell. As the authors show, this uncertainty is substantial and different individual cells have very different dose response. Since, it is not predetermined which of the cells may be involved in the actual response, this uncertainty should be taken into account when analyzing the capacity of signaling information transfer.
3. The analysis claims novelty in higher estimates of information transfer capacity. However, both study by Cheong et al. and Selimkhanov et al., provided the estimates of 1.5 bits or so for the signaling responses integrated over time. Since, as suggested above, the response analyzed in this study is adaptive and thus has the memory of the initial stimulation, this type of integrated response is also more appropriate for this study, and will likely yield the same result.
4. The study does not assess the complete signaling pathway(s). Calcium can stimulate multiple signaling outputs downstream and thus, unlike the NF-kappaB and MAPK signaling pathways analyzed by Cheong et al. and Selimkhanov et al., and many similar analysis afterwards, here the estimate of information transfer is incomplete. The information can be and is lost in subsequent steps in the signaling pathway and thus the estimate presented by authors can be misleading when compared to other studies.
5. The claim that the authors provide the lower bound in the information estimate is misleading. Indeed, capacity as a metric is in fact an upper bound on the amount of information that can be passed through an information channel (the signaling pathway in this case). The reason for the authors' claim is the sampling issues, but this issue can be dealt with in the standard way, presented for instance in Cheong et al. The same analysis results in the estimate of the confidence in the capacity measurement (expressed as an error bar).

Overall, these and many other concerns substantially lower my enthusiasm for this study and make

me recommend its rejection.

"High capacity in G protein-coupled receptor signaling", NCOMMS-17-04432

Reviewers' comments:

Reviewer #1 (Remarks to the Author):

Keshelava et al. present an evaluation of the information theory “channel capacity” concept in the context of a G-protein coupled receptor signaling system. Their experimental system is designed to observe single cells respond to multiple signaling events, in order to decouple responses from cell-to-cell variation. This is a very nice system that provides a new way to approach the question of channel capacity that has not been possible before, and the data obtained appear to be illuminating. The authors observe that individual cells can clearly respond differentially to gradations in stimuli, and they challenge previous claims of similar signaling systems being limited to binary discrimination of ligand presence. Overall, this is an elegant and technically sound experimental approach that is much more appropriate for making measurements of channel capacity than what has been done in previous work.

We are very thankful to the reviewer for this appraisal of the importance of our work and for the comments provided below, which we have addressed in our revision.

However, it is somewhat disappointing that despite this excellent system, the authors have limited themselves largely to following the form of the previous analyses (Cheong et al. for instance) and reporting a slightly higher channel capacity (1.65 bits vs 1 bit), rather than taking the chance to recast this model more substantially. I know many people who feel that the Cheong et al "1 bit" study is unconvincing and poorly executed, even if the general concept of measuring information capacity is interesting one. It would be a shame if readers of this work took away only the conclusion that cells can distinguish 3 levels of stimulus rather than 2, when in fact it may be possible to go quite a bit further.

Many thanks again! As detailed below, we have now added a theoretical framework to our paper, permitting to recalculate the channel capacity, from the lower bound of 1.65 bits provided in the initial

submission to 2.06 bits. Thus, cells are capable of distinguishing at least 4 different agonist concentrations using the GPCR signaling system under our study.

Major comments

1. It seems possible that a higher estimate of channel capacity could be made by altering the experimental setup. The authors note that the upper limit possible with 7 different doses is 2.8. However, for any given cell, most of the stimulant concentrations were saturating (either low or high), and it appears that few cells experienced more than 1 or 2 non-saturating doses, leaving their maximal observable capacity $\sim 1.5-2$. These limitations could be surpassed by tighter spacing of stimulant concentrations, spanning a suitable range for some cells to be sensitive. Theoretical extensions of the data could similarly offer estimates beyond the current limitations (see the following comment).

These are indeed very valuable suggestions. Given the different dynamic ranges different cells possess (see e.g. Fig. 3c) on the one hand, and the limited number of concentration points our setup could permit testing given the limited duration of the assay before any transcriptional feedback could be activated (see description on p.10) on the other, we could not experimentally tighten the spacing of the stimulant concentrations as this would force us to exclude many cells from the analysis, nor could we add many more concentrations to the ones tested. However, we took on the suggestion of the reviewer and expanded the concentration range *in-silico* by interpolating our measurements, as explained here and also in the answer to the next question of the reviewer.

We agree with the reviewer (and stated it in our initial submission) that the calculation of the channel capacity of 1.65 bits was a conservative lower bound estimate of the channel capacity. Following the reviewer's suggestion, we have in the revision investigated three different reasonable extensions of our analysis that go beyond this initial lower bound estimation. Those are described in response to the next comment.

2. The estimate of capacity appears to be based on a continuous error model for each cell at each concentration. However, the final estimate is performed with only the number of discrete concentrations used in the experiment. This propagates the fundamental limitation of the study, that relatively few stimulant concentrations may be used. However, the question of theoretical channel capacity over a continuous range of concentrations may still be addressed approximately. The response models at discrete concentrations could be extended to (smoothly)

estimate the response of each cell for any concentration, which may then be integrated to estimate the continuous channel capacity.

To continue the response to the previous comment, following the reviewer's suggestion, we first hypothesized that the cells whose dynamic response range coincides better with the agonist concentration range chosen in our experimental setup may reveal a higher channel capacity as calculated by our approach, than the other cells, whose internal dynamic range is shifted left- or right-wards in comparison to the experimental range of agonist concentrations. The cells in the first group will then show more non-saturating (either low or high) responses than the cells in the second group. Our analysis confirms the expectation, but only to a limited extent: we observed only a ca. 10% increase in the lower bound estimate of the channel capacity when restricting our analysis to cells with the optimal overlap of their dynamic range with the experimental agonist concentration range (new Supplementary fig. 1c).

Second, and in response to the comment of reviewer #2 on the extent of influence of adaptation in the signaling system (see below), we calculated the influence of this adaptation on our estimation of the channel capacity. As we expected given the low level of adaptation (with the response strength decreasing by ca. 1% to each consecutive cell stimulation), the influence of adaptation on the channel capacity calculation is rather limited: only 0.1 bit is "lost" in our channel capacity calculation due to adaptation.

And third, we have revised our analysis following the recommendation of the reviewer, which gave us a significant increase in the channel capacity estimate from 1.65 bits to 2.06 bits (see more below). Combining these three ways may further increase the calculation to ca. 2.3 bits; yet, we prefer to remain conservative and not mention this figure explicitly in our revised version of the manuscript, where we only state that our best estimate of 2.06 bit is still likely to be an underestimation.

Indeed, this suggestion of the reviewer to "(smoothly) estimate the response of each cell for any concentration" proved to be the most 'productive' way to go beyond the ultra-conservative calculation of the channel capacity provided in our initial submission. As illustrated by the new Fig. 3d and Supplementary Fig. 1b, and textual additions to the Results and Discussion and Methods, this interpolation approach allowed us justifying a recalculation of the channel capacity to be 2.06 ± 0.31 bits. As discussed above, even this calculation is likely an underestimation. This new estimate implies that individual cells in our system can reliably differentiate between at least four 4 agonist concentrations. We therefore believe that our work goes strongly beyond the previously published low estimates of channel capacity in intracellular signaling systems.

3. In discussing channel capacity, the definition of the "channel" for this study is only casually covered. The field stands to gain from careful consideration and discussion of how information theory concepts relate to real biological systems. In this case, the channel connects Ach outside

of the cell to free Ca^{++} in the cytoplasm. However, Ach and the associated GPCR pathway are certainly not the only factors that control free Ca^{++} - other factors may control the responsiveness of Ca^{++} levels to Ach via modification of receptors, cofactors, signal transducers, and their expression levels. These complexities confound the channel capacity concept, which (as it is used here) is derived for single-input/single-output channels. Since we are unable to simultaneously measure all relevant inputs to free Ca^{++} , any inputs other than Ach would appear as noise, though in actuality they do not detract from real multi-input/multi-output channel. It would be helpful to discuss where the current experimental system stands in excluding the unmeasurable effects of these additional inputs.

These are very important points to consider, and we are thankful to the review for inviting us to discuss them. In response to this suggestion, we have now added an extensive piece of text towards the discussion part of the paper (pp.12-13), as follows:

In a cell living in a complex environment and responding to a multitude of external and internal factors (e.g. cell-cell interactions, growth factors, cell growth and division cycle, etc.), each of the steps of the signaling pathway (channel) we studied may be influenced by these other factors in the form of expression level, post-translational modifications, and localization of the protein components, and production/release and degradation/removal of the second messenger components. Given the limited experimental control over these factors, they will be confounding for the channel capacity assessment of the signaling pathway and increase the noise in the measurements. These considerations further indicate that the high channel capacity we have calculated still underestimates the real capacity of intracellular signaling pathways.

The Ach-M3R-Gq-PLC β -IP $_3$ -Ca $^{2+}$ signaling pathway studied here in single HEK293 proceeds at levels below intracellular Ca $^{2+}$ to regulate multiple cellular activities, such as the activation of cellular kinases (e.g. different PKC isoforms) and their targets, cytoskeletal rearrangements, or transcription³³. We do not know whether bifurcation of the signal-transmitting channel downstream of intracellular Ca $^{2+}$ splits the high channel capacity into lower capacity sub-channels or whether the signaling pathway maintains the high capacity all the way down. Further, several of the intermediate components of the Ach-M3R-Gq-PLC β -IP $_3$ -Ca $^{2+}$ signaling system may have other effectors than those studied as the main signaling 'highway' in our work. As examples, the M3R GPCR can activate β -arrestins in addition of the heterotrimeric Gq protein²⁵; additional effectors of Gq-GTP and G $\beta\gamma$ released from the Gq heterotrimer exist in addition to PLC β ³⁴; and the second messenger IP $_3$ possesses multiple signaling outcomes in addition to opening intracellular Ca $^{2+}$ stores²⁸. It is thus clear that a signaling network, instead of a single isolated pathway, exists in cells and can be compared to a network of roads of different importance (capacity): highways and regional roads exiting from and entering to these highways at different points³⁵. In this analogy, the Ach-M3R-Gq-PLC β -IP $_3$ -Ca $^{2+}$ pathway we studied would represent a highway, whose channel capacity is measured as very high on the selected long distance. Channel capacity measurements of the subsequent parts of this road map and of the in- and out-coming regional routes should be a matter of subsequent studies, which would require the establishment of the experimental and theoretical framework permitting the application of information theory to a network of intracellular signal transduction.

4. It would also be helpful to consider and discuss other sources of variance, such as variance in each cell's actual exposure to Ach, pipetting errors, and noise in reporter measurements of Ca⁺⁺.

Our analysis shows, as we write in the paper (p.9), that “the response strength, being different in different individual cells, is highly reproducible within cells with a correlation $r = 0.999$. This value is higher than the $r = 0.9$ estimate obtained upon pairwise single-cell activation in another GPCR signaling system¹²”. Given this very high reproducibility (see Fig. 2c), the variance resulting from the limitations of the experimental setup (pipetting errors, any noise in the fluorescence reading, etc.) appears also to be minimal in our study. We have now added an additional comment relating to this on p.9: “This high reproducibility also indicates that any noise originating from the experimental imprecision (such as pipetting or fluorescent recording errors) is minimal in the experimental setup we constructed”.

Minor comments

1. Editing for language and punctuation errors is needed.

Every effort has now been made to correct the language and punctuation errors.

2. Various graph axis labeling etc.: Ensure suitable font size, inconsistent Y-axis labeling in Fig. 2B, missing concentration label in Fig. 2A,

We have now corrected these labeling issues.

3. Formatting of the methods “Channel capacity calculations” section appears troubled, especially with regard to equations. E.g. an extra appearance of $P(r,c)$ at line 359, and a missing parenthesis at line 362

The equations are now re-introduced in a proper format.

Reviewer #2 (Remarks to the Author):

The manuscript by Keshekava et al. addresses Ca signaling responses in response to stimulation of M3R receptors. The authors claim that the information transfer capacity of this signaling pathway is higher than previously reported, if one focuses on the single cell level dose response.

Unfortunately, in the opinion of this reviewer, the study, as presented, is misleading and the results and conclusions do not justify the messages presented. Below I present the reasons for this conclusion in no particular order.

1. The analysis treats the signaling system as essentially memoryless, thus assuming that the initial stimulus and response does not define the subsequent responses. However, the authors state themselves that the response is strongly adaptive, with each subsequent response decreasing in amplitude on the time scale of the experiment (for the same dose). In fact this is the reason the authors chose escalating doses in their analysis. This internal contradiction makes the analysis and the results unjustified.

Indeed, a ca.1% decrease in the response strength is seen in our measurements after a stimulation to each subsequent stimulation. We interpret this decrease as adaptation of the cell to stimulation. However, this decrease can hardly be considered strong. In this revision, we explicitly measure the influence of this adaptation on the calculated channel capacity (see p. 15 and Methods), coming to a conclusion that the adaptation decreases the channel capacity calculation by 0.1 bits. We thus confirm that the influence of the adaptation, as it exists in our experimental system, is a) tiny and b) decreases, rather than increases the estimate of the channel capacity.

We agree with the reviewer that the information theory with its concept of the channel capacity, as developed by Shannon, applies to memory-less information transmitting systems. Application of this theory to the intracellular signaling systems, which are capable to adapt, can thus serve only as an approximation, which, however, appears reasonable to us given the small influence of the adaptation we observed. Of note, the prior works on the application of the information theory to intracellular signaling ignored the issue of adaptability and provided no means to assess this adaptability and its influence on the calculated channel capacity.

In response to this point we now write in the discussion that subsequent developments may go in the direction of generalizing the channel capacity concept towards its more formal applications to adaptive cellular signaling systems (p.15):

“However, it may be insightful to move beyond the concept of channel capacity and consider the amount of information that can be transmitted by such adaptive systems, which is likely to further exceed the channel capacity. This would open interesting questions into adaptive information transmission in biological systems⁴⁰.

We believe that the experimental and theoretical framework we have provided in our paper will be very useful for such generalization and will advance further our understanding of the principles of intracellular signaling.

2. The authors misrepresent the prior studies and do not place their result in the context of the actual biological response. In fact, biologically, the uncertainty of the selection of the cells that are involved in response is as important if not important vs. the uncertainty of the dose response of an individual cell. As the authors show, this uncertainty is substantial and different individual cells have very different dose response. Since, it is not predetermined which of the cells may be involved in the actual response, this uncertainty should be taken into account when analyzing the capacity of signaling information transfer.

We agree with the reviewer that at the multicellular level, the variability of the response across cells adds a level of uncertainty, which has a biological meaning. However, while this uncertainty is likely to reduce the channel capacity of an ensemble of cells, it does not influence the channel capacity of a *single* cell, which is the focus of our study. Although different from population channel capacity, the study of single cell channel capacity is of fundamental importance for the intra-cellular signal transduction as it provides the basis from which to study further influences introduced at the population- or tissue-level.

3. The analysis claims novelty in higher estimates of information transfer capacity. However, both study by Cheong et al. and Selimkhanov et al., provided the estimates of 1.5 bits or so for the signaling responses integrated over time. Since, as suggested above, the response analyzed in this study is adaptive and thus has the memory of the initial stimulation, this type of integrated response is also more appropriate for this study, and will likely yield the same result.

The prior work of Cheong et al. did not provide analysis of integration of the response over time. In Cheong et al., a channel capacity above one was found when measuring not individual cells but cell ensembles (we now mention this in our Introduction, p.5). The study of Selimkhanov et al. relies on the same principle to study the channel capacity that was applied in the field before our work, i.e. single stimulation of cells with a fixed agonist concentration, followed by integration over many cells. As we discuss now in more detail in our Introduction, we believe this approach to assess the signaling channel capacity is limited. Indeed the higher estimate of 1.5 bits by Selimkhanov et al. requires the integration of the cellular response in individual cells over time. This is consistent with our finding, but not the same as the direct measurements of single cells with different concentrations that we performed in our study. In response to this comment we now better acknowledge the contribution of Selimkhanov et al., first in the

Introduction (pp.5-6: “Interestingly, extending the dimensionality of the readout by recording single cell dynamic responses to a single stimulation led to an estimated maximal mutual information between the ligand concentration and the (multidimensional) dynamic response to well above 1 bit¹⁵.”); and then again in the Results and Discussion (p.12: “Interestingly, our lower bound results are in line with the channel capacity of around 1.5 estimated from the single cell dynamic response published in the Selimkhanov et al. study, further validating its point that (given enough cells) the dynamic response partly compensates (in terms of mutual information) for the ignorance of the actual cell state in the estimation of the channel capacity.”).

4. The study does not assess the complete signaling pathway(s). Calcium can stimulate multiple signaling outputs downstream and thus, unlike the NF-kappaB and MAPK signaling pathways analyzed by Cheong et al. and Selimkhanov et al., and many similar analysis afterwards, here the estimate of information transfer is incomplete. The information can be and is lost in subsequent steps in the signaling pathway and thus the estimate presented by authors can be misleading when compared to other studies.

This is an important point, and we have added the following paragraph (already quoted above) to address this issue (pp. 13-14):

The Ach-M3R-Gq-PLCβ-IP₃-Ca²⁺ signaling pathway studied here in single HEK293 proceeds at levels below intracellular Ca²⁺ to regulate multiple cellular activities, such as the activation of cellular kinases (e.g. different PKC isoforms) and their targets, cytoskeletal rearrangements, or transcription³³. We do not know whether bifurcation of the signal-transmitting channel downstream of intracellular Ca²⁺ splits the high channel capacity into lower capacity sub-channels or whether the signaling pathway maintains the high capacity all the way down. Further, several of the intermediate components of the Ach-M3R-Gq-PLCβ-IP₃-Ca²⁺ signaling system may have other effectors than those studied as the main signaling ‘highway’ in our work. As examples, the M3R GPCR can activate β-arrestins in addition of the heterotrimeric Gq protein²⁵; additional effectors of Gq-GTP and Gβγ released from the Gq heterotrimer exist in addition to PLCβ³⁴; and the second messenger IP₃ possesses multiple signaling outcomes in addition to opening intracellular Ca²⁺ stores²⁸. It is thus clear that a signaling network, instead of a single isolated pathway, exists in cells and can be compared to a network of roads of different importance (capacity): highways and regional roads exiting from and entering to these highways at different points³⁵. In this analogy, the Ach-M3R-Gq-PLCβ-IP₃-Ca²⁺ pathway we studied would represent a highway, whose channel capacity is measured as very high on the selected long distance. Channel capacity measurements of the subsequent parts of this road map and of the in- and out-coming regional routes should be a matter of subsequent studies, which would require the establishment of the experimental and theoretical framework permitting the application of information theory to a network of intracellular signal transduction.

Having added these considerations, we would like to stress that the prior works, calculating the channel capacity as ca. 1 bit, arrived to this conclusion not only looking at the very downstream cellular responses, but also measuring the same response as we did, i.e. release of intracellular Ca^{2+} (Cheong et al; Selimkhanov et al.). Further, MAPK activation measured e.g. in Selimkhanov et al. cannot be clearly considered more 'downstream' in the receptor tyrosine kinase signaling (containing a chain EGF-EGFR-Sos-Ras-ERK with five intermediates) than that of Ca^{2+} in the Ach-M3R-Gq-PLC β -IP $_3$ - Ca^{2+} signaling system.

The reviewer seems to propose that below Ca^{2+} , the channel capacity should be reduced. As we write in the paragraph cited above, this is in fact not known and will require future studies, applying the approach developed in our work, for the proper elucidation.

5. The claim that the authors provide the lower bound in the information estimate is misleading. Indeed, capacity as a metric is in fact an upper bound on the amount of information that can be passed through an information channel (the signaling pathway in this case). The reason for the authors' claim is the sampling issues, but this issue can be dealt with in the standard way, presented for instance in Cheong et al. The same analysis results in the estimate of the confidence in the capacity measurement (expressed as an error bar).

We understand the confusion that may arise in the mind of the reader by talking about the lower bound on channel capacity, which is itself a higher bound. Indeed, we realize that estimating a "lower bound of a higher bound" (which is what we did) may not be the most intuitive. This, together with comments from the first reviewer, leads us to directly estimate the channel capacity by interpolating the cell response distribution, and only mention the lower bound incidentally.

Overall, these and many other concerns substantially lower my enthusiasm for this study and make me recommend its rejection.

We can only hope that the reviewer may reconsider his/her estimation of our paper, taking into consideration our revision, the responses to the reviewers' comments, and the comments of the first reviewer.

Reviewers' comments:

Reviewer #1 (Remarks to the Author):

We feel that the authors have adequately addressed all of our concerns. We are in favor of publication for the revised version of the manuscript.

Reviewer #2 (Remarks to the Author):

In the revised manuscript, Kshelava et al., made mostly stylistic revisions of the text, attempting to address my comments either directly or by making relatively minor changes in the text. One exception is the analysis of the role of adaptation in the signaling pathway, something that is addressed below. Sadly, these clarification do little to address my initial concerns, and make me continue to believe that the study, although potentially interesting, will only increase the confusion about the notion of information processing in biochemical pathways, without helping to develop it. As, arguably, calculation of information capacity is the main goal of this study (other biological advances are minor and not novel), this point therefore continues to be a key to my evaluation of the study. Below, I will confine myself to the particular questions raised in the first round of review and the responses of the authors to my comments:

A general comment. The authors continue to present this study in opposition to that of Cheong et al., and multiple other studies that followed this initial analysis of the capacity of biochemical pathways. Although the authors softened this stance in the response to the reviewer, it continues to be a strong claim of the study. I continue to believe that this way to present their study is very misleading. Indeed, the manuscript suggests that a given cell can have very different response profile than another cell. In particular, it may be not be responsive to concentrations that another cell can be responsive to. This suggests that, in biological terms, here, a lot depends on which cell is involved in the response. It is therefore not the analysis of a generalized signaling pathway but rather of a cell chosen from a very diverse population. This choice, in information theoretical terms, means that the uncertainty of the response is reduced, and thus, the information capacity is increased, by the researchers themselves. This additional information increase may account for the increase of approximately 0.5 bits vs. prior estimates reported elsewhere. This is the key point distinguishing the current study from prior ones, and thus, the study is in no way contradictory to prior ones, and thus should not be presented as such. This, coupled with multiple issues with analysis that I focus on below, makes me believe that this study continues to misrepresent the analysis of the capacity of signaling in biochemical pathways.

1. The consistency/adaptive nature of cell response. Although the authors attempted to address this question in the revision, suggesting that the response amplitude may drop by 1% following a stimulation by the same input, the effect is not as small as it is represented to be, as each experiment involved multiple repeated stimulations, and the accumulated adaptive effect was very considerably during the final rounds of stimulation. The adequate way to address this point would be to expose the cells not to escalating doses of input but to a randomized sequence of doses (which is also a potentially more biologically relevant scenario). Unfortunately, this was not done leaving the question of the role of adaptation to the input still open. Given this, it was especially striking to see the authors claim that the effect of adaptation actually leads to underestimation of the signaling information capacity. Adaptive response actively limits the ability to distinguish between different stimuli and therefore cannot suggest a higher than estimated information capacity, only a lower one.

Furthermore (see below on that), in Fig. 2b and subsequent analysis, the authors clearly very inappropriately and misleadingly, have selected three examples of the cells showing close to saturating response at 250 nM dose. As can be seen in Fig. 3a, many cells either do not respond at this doses or show much more variable response (for them this does is far from saturation). Thus the analysis of the variability at this saturating dose is very misleading and cannot be used to

justify the statements in the analysis.

2. Related to the previous point of cell response to different subsequently presented doses. The analysis, as illustrated in Fig. 3a, suggests that the response to intermediate doses is far less consistent than implied by the analysis in Fig. 2. Indeed, this figure suggests that the response saturates almost immediately, after just two dose escalations, raising further serious doubts about the capacity analysis precision. More importantly, at the doses prior saturating ones, the response amplitude of individual peaks shows a high degree of uncertainty. It either gradually escalates (case 2) with each step change being far in excess of 1% claimed above, or widely oscillates (case 1), or appears random (case 3). This suggests that the response is far less consistent than is suggested by the analysis and thus far less predictable for the cell vs. the claims made. Indeed, the analysis suggests that the cells, at doses used, are either not responding, or responding at saturation, or responding at a highly variable level that cannot provide the a valuable estimate of the external input level. Given this and other examples, I cannot possibly see how the capacity value can be as high as 2 bits (reliable evaluation of 4 distinct input levels).

2. Evaluation of information capacity. as shown by Cheong et al., the estimate of the information capacity depends on such key analysis details as data binning, etc. This is discussed extensively in the Supp. Materials of that paper, specifying in particular how the information estimate confidence interval can be obtained. This key analysis continues to be omitted in this study. This really important, since (as clearly shown in examples in Fig. 3), cells effectively only respond distinctly and reliably within a very narrow range of input doses, much narrower than the range of inputs used by the authors.

3. The authors continue to misrepresent the results by Cheong et al. They continue to claim that that study did not evaluate single cell responses and that it did not study integration of over time. On the contrary, the analysis there explicitly showed that integration over time can occur, and measured it directly by using GFP reporters of NF-kappaB transcriptional regulation. GFP accumulation over extended periods of time was explicitly analyzed and interpreted. As explicitly shown in the last Supplementary Figure of that study, the time integration in this pathway can lead to an increase in the channel capacity to approximately 1.5 bits. It is not clear to me why the authors choose to explicitly counterclaim this point. Again, I believe that the analysis in this current manuscript does not suggest any higher estimate than that already presented in Cheong et al., and Selmkhanov et al., studies.

4. Although it is good to see the authors to attempt to address the problem that they did not evaluate the capacity of the whole signaling pathway, but rather at the level of calcium output, the new discussion is rather confusing. I do not understand the analogy to a network of roads and highways. It is extremely confusing. Again, the point here is that Calcium cannot be seen as the final point of a signaling process, as e.g., NF-kappaB or activation of other transcription factors can be. Rather Calcium is an intermediate point of many signaling pathways, followed by multiple subsequent steps, activating Calcium-dependent signaling processes. Again, by the nature of the fundamental theorem of information theory, information can only decrease as it propagates down the signaling pathways. To use the very unfortunate analogy by the authors, events leading to Calcium activity represent an entrance point to a highway, not the highway (or highway network) itself. This is a key point for comparison of their results to results of a multitude of other studies, where the signaling pathway analyses are much more complete.

Overall, I continue to believe that the study is misleading as it cannot convincingly claim that for a cell randomly selected (either within an actual biological process or by the authors experimentalist in this study) has the ability to reliably distinguish between doses in the range pre-selected by the authors with the capacity as high as 2 bits (4 doses). Even if the cells can integrate their measurements over time (which is an unproven assumption) and are exposed to different, sequentially presented doses (another unproven assumption), their ability to do so is more likely is close to 1-1.5 bits estimates made in a host of previous studies. This ability is clearly compromised

due to differential sensitivity of the cells to the fixed dose range used, variability of their response at intermediate dose ranges and adaptive nature of their responses at saturating doses. As such, I think study, although providing another example of a sensory response analyzed using information theory, does not increase our understanding of information processing in biochemical pathways and networks, while also having a high potential to present confusing and misleading claims. Therefore, I cannot support its publication in this journal.

Point-by-point responses to the Reviewers' comments. Our responses are provided in blue below each individual comment of the reviewers.

Reviewer #1 (Remarks to the Author):

We feel that the authors have adequately addressed all of our concerns. We are in favor of publication for the revised version of the manuscript.

We are very thankful for Reviewer #1 for his/her assessment of our manuscript, both in its previous form, and now in its revised form.

Reviewer #2 (Remarks to the Author):

In the revised manuscript, Kshelava et al., made mostly stylistic revisions of the text, attempting to address my comments either directly or by making relatively minor changes in the text. One exception is the analysis of the role of adaptation in the signaling pathway, something that is addressed below. Sadly, these clarification do little to address my initial concerns, and make me continue to believe that the study, although potentially interesting, will only increase the confusion about the notion of information processing in biochemical pathways, without helping to develop it. As, arguably, calculation of information capacity is the main goal of this study (other biological advances are minor and not novel), this point therefore continues to be a key to my evaluation of the study. Below, I will confine myself to the particular questions raised in the first round of review and the responses of the authors to my comments:

A general response to the comments of Reviewer #2. We feel that the Reviewer, being in general opposed to our work, in his/her detailed comments to this revised version of our manuscript demonstrates a strong bias, being often unjustified in his/her assessment. We will detail these examples below. In this introductory statement, the Reviewer writes that our revision is mostly stylistic. This completely ignores the fact that, in the revision, we have added an important mathematical expansion to our experimental and theoretical work, following the recommendations of Reviewer #1. This expansion led to the overall calculation of the channel capacity in individual cells being above 2.0 bits – by far exceeding all prior estimations and going against some of the comments Reviewer #2 provided in his/her previous review.

A general comment. The authors continue to present this study in opposition to that of Cheong et al., and multiple other studies that followed this initial analysis of the capacity of biochemical pathways. Although the authors softened this stance in the response to the reviewer, it continues to be a strong claim of the study. I continue to believe that this way to present their study is very misleading. Indeed, the manuscript suggests that a given cell can have very different response profile than another cell. In

particular, it may be not be responsive to concentrations that another cell can be responsive to. This suggests that, in biological terms, here, a lot depends on which cell is involved in the response. It is therefore not the analysis of a generalized signaling pathway but rather of a cell chosen from a very diverse population. This choice, in information theoretical terms, means that the uncertainty of the response is reduced, and thus, the information capacity is increased, by the researchers themselves. This additional information increase may account for the increase of approximately 0.5 bits vs. prior estimates reported elsewhere. This is the key point distinguishing the current study from prior ones, and thus, the study is in no way contradictory to prior ones, and thus should not be presented as such. This, coupled with multiple issues with analysis that I focus on below, makes me believe that this study continues to misrepresent the analysis of the capacity of signaling in biochemical pathways.

In this general comment, the Reviewer brings up two issues, which need to be addressed separately. The first is that, in the eyes of the Reviewer, our manuscript is written in the way opposing some prior studies in this field. We have attempted to soften this opposition in our revision, concentrating on the fact that our approach to estimate the individual cells' channel capacity is experimentally different from the approaches taken in the prior studies.

However, we disagree with the second point the Reviewer brings up here. In our study, as is explicitly described in the main text and in the Methods, there is no preselection of cells from the population treated with the agonist. Within the population, different cells indeed have individual 'windows' of operation, responding differently to different agonist concentrations. Yet every cell within the population is analyzed in terms of the response and the resulting channel capacity, and the Reviewer and readers are able to assess the responses and the calculations regarding each of the hundreds of the cells we analyzed through the supplementary information. Thus, the channel capacity of the cells as we provide it is not increased by the researchers.

1. The consistency/adaptive nature of cell response. Although the authors attempted to address this question in the revision, suggesting that the response amplitude may drop by 1% following a stimulation by the same input, the effect is not as small as it is represented to be, as each experiment involved multiple repeated stimulations, and the accumulated adaptive effect was very considerably during the final rounds of stimulation.

It is unfortunate that here, the Reviewer preferred his/her impressions to the direct calculations provided in our work. On average, each individual stimulation decreases the response amplitude to the subsequent stimulation by 1% (line 173, p.9). Over 20 rounds of stimulation with the same concentration, the decrease in the response amplitude is ca. 18% (line 174, p.9). In our main experimental setup, cells are stimulated with 5 pulses of 7 different concentrations, given in the escalating dosage (100nM, 250nM, 500nM, 750nM, 1.5 μ M, 3 μ M, and 10 μ M) to minimize the impact of desensitization (lines 202, p.10). In this protocol of 5 repeated stimulations, no desensitization was seen for agonist concentrations below 750nM (Supplementary Fig. 1, see also text: lines 177-179, p.9). The final effect of the desensitization in our experimental protocol on the outcome of the calculated channel capacity is an underestimation of the channel capacity by 6% (line 300, p.15 of the main text; also see

the legend to Supplementary Figure 1, and the script for providing the analysis given as Supplementary Data 4).

The adequate way to address this point would be to expose the cells not to escalating doses of input but to a randomized sequence of doses (which is also a potentially more biologically relevant scenario). Unfortunately, this was not done leaving the question of the role of adaptation to the input still open.

The reasons to performing the escalating course of stimulations are explicitly given in the text (lines 201-204, pp.10-11). Doing the random stimulations is no more biologically relevant than doing the escalating course, as in the real life cells may not be faced repeatedly with different agonist concentrations on such a short scale anyway. It is clear to readers and the Reviewer that we here, as the researchers applying information theory to intracellular signaling before us, designed an artificial experimental setup to probe the cell's intrinsic transducing capacity. Thus hints towards lower or higher biological relevance in this regard should be avoided. As detailed in our response above, the question of the role of adaptation to the input is well addressed.

Given this, it was especially striking to see the authors claim that the effect of adaptation actually leads to underestimation of the signaling information capacity. Adaptive response actively limits the ability to distinguish between different stimuli and therefore cannot suggest a higher than estimated information capacity, only a lower one.

We would prefer the Reviewer to directly look at our calculations, instead of choosing general considerations. The readers can do this, running the R script we provide as Supplementary Data 4 on the raw data we provide as the other Supplementary Data files. The calculation is simple and shows that, when the adaptation (sensitization) of the cells' responses is corrected for (Supplementary Fig. 1), the resulting channel capacity calculation is increased by 6%.

Intuitively, this effect is very simple to understand. Imagine that a cell is stimulated by the agonist at two different concentrations, the second being somewhat higher than the first one. If no sensitization happens, the second cell's response to the second stimulus is on average higher than the first one. This observation will result in a certain calculation of the channel capacity x . If, in contrast, due to sensitization the second stimulation produces (on average) the same response as the first stimulation, this will decrease the resulting channel capacity y , ($y < x$).

In our calculation of the channel capacity, we chose to stay on the more conservative side, providing the final calculation of channel capacity as y (which equals 2.06 bits, line 221, p.11). If, however, we compensate for the sensitization as observed in our experimental protocol, the channel capacity calculation would increase by 6%, to 2.18 bits.

Furthermore (see below on that), in Fig. 2b and subsequent analysis, the authors clearly very inappropriately and misleadingly, have selected three examples of the cells showing close to saturating response at 250 nM dose. As can be seen in Fig. 3a, many cells either do not respond at this doses or show much more variable response (for them this does is far from saturation). Thus the analysis of the

variability at this saturating dose is very misleading and cannot be used to justify the statements in the analysis.

We are very disappointed by the negligence in the analysis of our data the Reviewer has demonstrated here. Figure 2b shows responses of three different cells to 20 repeats of stimulation with the same concentration (250nM) of the agonist. The purpose of Fig.2b is to show exactly the opposite to what the Reviewer has seen – that different cells respond at different levels to the same agonist concentration, with the cell on the top responding at ca. 0.3 response units, cell in the middle – at ca. 0.25 units, and cell in the bottom – at ca. 0.15 units. It suffices to look then at the panel 2c in order to check what are different cells' responses to the stimulation at 250nM agonist, where not just three cells are given but dozens. It is clear that, indeed, some cells respond very little to this concentration (as two cells shown at Fig. 3a, indeed), but the vast majority did respond, with the level of response varying from ca. 0.15 to 0.3 – just as the three representatives selected for the panel 2b. More examples can be seen on panel 2a, on the Supplementary Movie, and, finally, in the Supplementary Data files, where each of the >400 hundred cells analyzed in our work are given. We must unfortunately conclude that the Reviewer chose to close the eyes and not see what is clearly provided in our manuscript.

2. Related to the previous point of cell response to different subsequently presented doses. The analysis, as illustrated in Fig. 3a, suggests that the response to intermediate doses is far less consistent than implied by the analysis in Fig. 2. Indeed, this figure suggests that the response saturates almost immediately, after just two dose escalations, raising further serious doubts about the capacity analysis precision. More importantly, at the doses prior saturating ones, the response amplitude of individual peaks shows a high degree of uncertainty. It either gradually escalates (case 2) with each step change being far in excess of 1% claimed above, or widely oscillates (case 1), or appears random (case 3). This suggests that the response is far less consistent than is suggested by the analysis and thus far less predictable for the cell vs. the claims made. Indeed, the analysis suggests that the cells, at doses used, are either not responding, or responding at saturation, or responding at a highly variable level that cannot provide the a valuable estimate of the external input level. Given this and other examples, I cannot possibly see how the capacity value can be as high as 2 bits (reliable evaluation of 4 distinct input levels).

This paragraph is another excellent example of the attitude of the Reviewer, preferring general considerations to the actual and careful reading of our paper. Fig. 2 does not imply any analysis, but *provides*, in an unbiased manner, results of the response of dozens of cells to repeated stimulations with one selected concentration. Fig. 3a (and 3c) gives examples of some selected cells (three cells on Fig. 3a, 4 cells of Fig. 3c). The cells of the panel 3a were selected in order to illustrate the fact that some cells respond well only to high concentrations, while the others – already to the low ones. These cells do show the intermediate responses, and these intermediate responses are illustrated further at the panel 3c. The Reviewer could have also looked at the >400 individual cells provided by the Supplementary Data, and run the script provided in the Supplementary Data 4.

2. Evaluation of information capacity. as shown by Cheong et al., the estimate of the information capacity depends on such key analysis details as data binning, etc. This is discussed extensively in the

Supp. Materials of that paper, specifying in particular how the information estimate confidence interval can be obtained. This key analysis continues to be omitted in this study. This really important, since (as clearly shown in examples in Fig. 3), cells effectively only respond distinctly and reliably within a very narrow range of input doses, much narrower than the range of inputs used by the authors.

Our way of estimating channel capacity differs from Cheong et al. because we use a parametric approach, whereas their approach is non-parametric. The parametric approach is probably more powerful but makes some assumptions on the data, while the non-parametric approach requires a binning procedure, which is non-trivial. So it is not that we omitted the key analysis, we replaced it by another procedure, which is more powerful.

3. The authors continue to misrepresent the results by Cheong et al. They continue to claim that that study did not evaluate single cell responses and that it did not study integration of over time. On the contrary, the analysis there explicitly showed that integration over time can occur, and measured it directly by using GFP reporters of NF-kappaB transcriptional regulation. GFP accumulation over extended periods of time was explicitly analyzed and interpreted. As explicitly shown in the last Supplementary Figure of that study, the time integration in this pathway can lead to an increase in the channel capacity to approximately 1.5 bits. It is not clear to me why the authors choose to explicitly counterclaim this point. Again, I believe that the analysis in this current manuscript does not suggest any higher estimate than that already presented in Cheong et al., and Selmkhanov et al., studies.

We chose to discuss separately the 'time integration' procedure of Selimkhanov et al. and that of Cheong et al, because the one of Selimkhanov et al. put considerable attention not just to the accumulation of a target protein (GFP) over time, but to the kinetics of cellular response. But we agree that already in the Cheong et al. paper, the issue of integration over time was provided. For the sake of avoiding misinterpretation of what we intended to mean, in this second revision we now put together the references for Selimkhanov et al. and Cheong et al when discussing the prior studies on the time integration.

4. Although it is good to see the authors to attempt to address the problem that they did not evaluate the capacity of the whole signaling pathway, but rather at the level of calcium output, the new discussion is rather confusing. I do not understand the analogy to a network of roads and highways. It is extremely confusing. Again, the point here is that Calcium cannot be seen as the final point of a signaling process, as e.g., NF-kappaB or activation of other transcription factors can be. Rather Calcium is an intermediate point of many signaling pathways, followed by multiple subsequent steps, activating Calcium-dependent signaling processes. Again, by the nature of the fundamental theorem of information theory, information can only decrease as it propagates down the signaling pathways. To use the very unfortunate analogy by the authors, events leading to Calcium activity represent an entrance point to a highway, not the highway (or highway network) itself. This is a key point for comparison of their results to results of a multitude of other studies, where the signaling pathway analyses are much more complete.

This part of the discussion was added as asked by Reviewer #1, and he/she appears to be very happy with it. The analogy to roads is taken from the insightful discussions with Marc Kirschner, a leading systems pharmacologist (the citation #35 is provided for this). In any case, this metaphor is in no way central to the work, and I think it is clear to the readers that we are providing some illustrative discussions here.

As for the issue of comparing Ca^{2+} and NF-kappaB, I wish to copy our reply to this comment of this reviewer we had provided in our initial revision:

Having added these considerations, we would like to stress that the prior works, calculating the channel capacity as ca. 1 bit, arrived to this conclusion not only looking at the very downstream cellular responses, but also measuring the same response as we did, i.e. release of intracellular Ca^{2+} (Cheong et al; Selimkhanov et al.). Further, MAPK activation measured e.g. in Selimkhanov et al. cannot be clearly considered more 'downstream' in the receptor tyrosine kinase signaling (containing a chain EGF-EGFR-Sos-Ras-ERK with five intermediates) than that of Ca^{2+} in the Ach-M3R-Gq-PLC β -IP β - Ca^{2+} signaling system.

The reviewer seems to propose that below Ca^{2+} , the channel capacity should be reduced. As we write in the paragraph cited above, this is in fact not known and will require future studies, applying the approach developed in our work, for the proper elucidation.

Overall, I continue to believe that the study is misleading as it cannot convincingly claim that for a cell randomly selected (either within an actual biological process or by the authors experimentalist in this study) has the ability to reliably distinguish between doses in the range pre-selected by the authors with the capacity as high as 2 bits (4 doses). Even if the cells can integrate their measurements over time (which is an unproven assumption) and are exposed to different, sequentially presented doses (another unproven assumption), their ability to do so is more likely is close to 1-1.5 bits estimates made in a host of previous studies. This ability is clearly compromised due to differential sensitivity of the cells to the fixed dose range used, variability of their response at intermediate dose ranges and adaptive nature of their responses at saturating doses. As such, I think study, although providing another example of a sensory response analyzed using information theory, does not increase our understanding of information processing in biochemical pathways and networks, while also having a high potential to present confusing and misleading claims. Therefore, I cannot support its publication in this journal.

The channel capacity of a given cell cannot depend on the response of other cells, and therefore it cannot depend on "differential sensitivity of the cells to the fixed dose range used". The fact that we, as observers, cannot distinguish cells with different sensitivity does not affect their channel capacity. The only thing this can effect (and which can be discussed) is the biological relevance of knowing the channel capacity of such indistinguishable cells.

Overall, we must conclude that, unfortunately, the Reviewer has displayed an unjustifiable degree of bias in analyzing our work, closing his/her eyes to the obvious and misinterpreting our data and discussions at several places. It is clear that the Reviewer does not wish our work to be published, as it contradicts his/her previous publications. However, I am also certain that such stance is far from being scientific.

Reviewers' Comments:

Reviewer #3:

Remarks to the Author:

In their manuscript, Keshelava et al. use single-cell responses to stimulus dose-escalation to estimate the single-cell channel capacity of a GPCR pathway, namely the 'Ach-M3R-Gq-PLC β -IP3-Ca $^{2+}$ ' signaling axis. This is an important and timely question which is well delineated in Figure 1. They find that, on average, cells distinguish around 4 concentrations of acetylcholine, a channel capacity of ~ 2 . Their work is appropriately placed in the context of a recent set of studies evaluating the channel capacity of a few different signaling pathways, some suggesting that, at best, individual cells can evaluate whether a signal is "on" or "off", while other suggesting that individual cells may be able to resolve more finely the strength of a stimulus.

Suggestions for minor text revisions:

P.12 lines 248-250 – the authors may want to be even more explicit and direct in writing out the reasoning behind how their findings validate how dynamic responses partly compensates for the ignorance of the actual cell state.

P.13 lines 259-260 – the authors state that "the high channel capacity we have calculated still underestimates the real capacity of intracellular signaling pathways." – with reference to the fact that uncontrolled variability in the abundance of molecular players in the pathway may overall degrade the assessed channel capacity. This is one point that may require further nuance or a more explicit reasoning: as each instantiation of the system is observed, in theory the estimated capacity observed is its true capacity. However, because this variability affects the absolute concentration range to which a particular cell matters, and because each cell is examined in its response to a fixed set of Ach concentrations, the capacity of some cells (particularly sensitive, or particularly insensitive) may be underestimated. This is discussed in the methods section, but for greater clarity it should be either referred to or repeated in the main text.

I understand that the manuscript has been revised in response to prior reviews. I have read the authors' detailed response to the concerns expressed by Reviewer 2. On most points, I agree with the authors. In my opinion, they have carefully defined their question and appropriately positioned their work in the existing body of literature. Under this framework, their interpretations of their data and analyses are overall well-supported and suitably nuanced and therefore I am in favor of publication of this revised manuscript.

- The careful characterization of the memory phenotype is interesting and new, especially in the context of its potential impact on channel capacity
- The study is appropriately situated with regards to previously published analyses of information capacity in signal transduction pathways, and the specific question investigated is very clearly delineated by Figure 1
- Point 1 – The size of the effect of the adaptative behavior is clearly described in the main text and used to nuance interpretations.
- In my opinion the physiological relevance of the stimulation format is a moot point. Here the purpose of the stimulation is to characterize the underlying system, and for some of these characterizations a non-physiological stimulus can be much more informative than a physiological stimulus. The authors clearly justify their choice in the text. Although it may have been interesting to study the response to other forms of repeated stimulation, it is more a matter of scientific curiosity, than a matter of rigor of analysis.
- The argument about whether the adaptative nature of the cellular response leads to an underestimation of channel capacity seems to be one of semantics. If I understand correctly, whereas the reviewer seems to consider the *observed* response as the *effective* response. The authors address a 'what-if' scenario by calculating a correction – answering what may be the theoretical capacity of the system, if there were no adaptation. This seemed to me sufficiently clear in the main text of the manuscript.
- Point 2. The authors provide sufficient details about their analyses to allow readers to make their own judgement
- Point 3. The message that was emphasized in Cheong et al was the low channel capacity, and so the background literature seems appropriately situated, and additionally, the authors do allow a

nuance to this message and now reference a discussion of dynamics in Cheong et al. in the revised text.

- Point 4. The analogy may help some readers, and the manuscript clearly defines which pathways capacities are estimated in their work and how they relate to other pieces of the signaling network.

- I believe that the authors have carefully defined what system they investigated and which channel capacity they have estimated. Readers should be able to position this work in the context of other assessments of channel capacity and therefore the work does not present misleading claims.

Reviewer #4:

None

Reviewer #5:

Remarks to the Author:

The manuscript by Keshelava et al. claims to have uncovered a capacity for GPCR to transmit signals above binary level formally designated as having channel capacity of more than 2. Experimental evidence to support this claim comes from analysis of intracellular Ca^{2+} responses in HEK293 cells to addition of acetylcholine presumed to activate the M3 receptor. The conclusions come from the analysis of concentration-dependence of responses recorded from individual cells. The data are then fed into a set of equations to model it in terms of information theory from which it is concluded that GPCRs are capable of signaling at channel capacity of more than one basically implying that they are more than on/off switches. While this reviewer cannot judge the validity or impact of the theoretical work and its value to the field of computational modeling, from the biological perspective the premise appears to be highly superficial, the experimental results are entirely predictable and documented in hundreds of papers and implications for understanding cellular signaling quite dubious. Specific points are as follows.

1. The major technological advance here is touted as the first ever analysis of GPCR responses on the single cell level. This is simply not true. Hundreds of studies examined single cell responses of individual cells to GPCR stimulation using by far more precise and interpretable method of patch clamp electrophysiology. This includes classical work on light stimulation of vertebrate and invertebrate photoreceptors, recording of K^+ channel responses in cardiomyocytes and neurons as well as neurotransmitter modulation of the voltage sensitive Ca^{2+} channels among many others. All of these classical studies used endogenous physiological systems that are by far more defined than Ca^{2+} responses in HEK cells. Furthermore, mathematical modeling of the data had been performed and widely available.

2. It is widely known and again validated across multiple systems, including work mentioned that GPCRs exhibit gradual responses at single cell level in a dose-dependent fashion, so theoretical motivation and alternatives presented in Figure 1 are entirely misleading. GPCR cascades do not function as binary switches.

3. Based on textbook knowledge the signaling capacity of GPCRs systems should be far beyond 2. The operational model of GPCR introduces the bias factor in addition to potency and efficacy signaling (see reviews by Kenakin) which already make the system pluridimensional. The authors appear to ignore efficacy dimension by normalizing their data to % max. Furthermore, GPCRs use multiple transducers for signaling, e.g. beta arrestins and G proteins which further increases channel capacity- looking only at one readout ignores this complexity.

4. In addition to the considerations above, analyzing Ca^{2+} response as a proxy for GPCR signaling is far too downstream for being able to conclude how

Reviewer #3 (Remarks to the Author):

In their manuscript, Keshelava et al. use single-cell responses to stimulus dose-escalation to estimate the single-cell channel capacity of a GPCR pathway, namely the 'Ach-M3R-Gq-PLC β -IP3-Ca $^{2+}$ ' signaling axis. This is an important and timely question which is well delineated in Figure 1. They find that, on average, cells distinguish around 4 concentrations of acetylcholine, a channel capacity of ~ 2 . Their work is appropriately placed in the context of a recent set of studies evaluating the channel capacity of a few different signaling pathways, some suggesting that, at best, individual cells can evaluate whether a signal is "on" or "off", while other suggesting that individual cells may be able to resolve more finely the strength of a stimulus.

We sincerely thank the reviewer for the overall positive assessment of our work, and provide the responses to each of the minor issues below.

Suggestions for minor text revisions:

P.12 lines 248-250 – the authors may want to be even more explicit and direct in writing out the reasoning behind how their findings validate how dynamic responses partly compensates for the ignorance of the actual cell state.

We have expanded this section, following the suggestion of the reviewer to be more explicit and direct. Now this section reads as follows:

Interestingly, our lower bound results are in line with the channel capacity of around 1.5 estimated from the single cell dynamic response published in the Selimkhanov et al. study¹⁵, further validating its point that (given enough cells) the dynamic response partly compensates (in terms of mutual information) for the ignorance of the actual cell state in the estimation of the channel capacity. In other words, measuring single cell responses to single stimuli at multiple time points indeed helps to some extent distinguish between extrinsic (i.e. across cells) and intrinsic (i.e. within cells) response variability.

P.13 lines 259-260 – the authors state that "the high channel capacity we have calculated still underestimates the real capacity of intracellular signaling pathways." – with reference to the fact that uncontrolled variability in the abundance of molecular players in the pathway may overall degrade the assessed channel capacity. This is one point that may require further nuance or a more explicit reasoning: as each instantiation of the system is observed, in theory the estimated capacity observed is its true capacity. However, because this variability affects the absolute concentration range to which a particular cell matters, and because each cell is examined in its response to a fixed set of Ach concentrations, the capacity of some cells (particularly sensitive, or particularly insensitive) may be underestimated. This is discussed in the methods section, but for greater clarity it should be either referred to or repeated in the main text.

We have expanded this section, following the suggestion of the reviewer to be more explicit. Now this section reads as follows:

Given the limited experimental control over these factors, they will be confounding for the channel capacity assessment of the signaling pathway and increase the noise in the measurements. This variability further affects the concentration range to which a particular cell responds, 'shifting' this range for some cells away from the range fixed in the experiment. These considerations further indicate that the high channel capacity we have calculated still underestimates the real capacity of intracellular signaling pathways.

Reviewer #5 (Remarks to the Author):

The manuscript by Keshelava et al. claims to have uncovered a capacity for GPCR to transmit signals above binary level formally designated as having channel capacity of more than 2. Experimental evidence to support this claim comes from analysis of intracellular Ca²⁺ responses in HEK293 cells to addition of acetylcholine presumed to activate the M3 receptor. The conclusions come from the analysis of concentration-dependence of responses recorded from individual cells. The data are then fed into a set of equations to model it in terms of information theory from which it is concluded that GPCRs are capable of signaling at channel capacity of more than one basically implying that they are more than on/off switches. While this reviewer cannot judge the validity or impact of the theoretical work and its value to the field of computational modeling, from the biological perspective the premise appears to be highly superficial, the experimental results are entirely predictable and documented in hundreds of papers and implications for understanding cellular signaling quite dubious.

It seems that, unfortunately, the reviewer has not looked at our work in the context of the field of application of information theory to intracellular signaling and of the current knowledge in this field. Experimentation on intracellular signal transduction indeed dates back for decades. Information theory, being a branch of math analyzing passage, storage, and encoding of information, however, has been applied to biological systems very scarcely, and to intracellular signaling – not until very recently. All the papers dealing with these applications are cited and discussed in the Introduction and subsequent sections of our paper. These prior studies paradoxically came to a conclusion that the channel capacity of the intracellular signaling pathways equals one bit, i.e. that a cell can transmit only an “on-off” signal, not being able to discriminate between different levels of activation. Being of a fundamental importance to signal transduction, this fundamental claim, as we show, has not been achieved through the proper experimental approach. We design the experimental paradigm, which allows the proper measurement of the individual cell's channel capacity, and show that this channel capacity is high, i.e. that the cells' signal transduction systems are able to reliably transmit information on at least four different levels of the extracellular stimulus. Thus, provided the domineering “1 bit” opinion in the field, supported by several high-impact publications, our work provides both a fundamental advance into the basic principles of cellular signaling, as well as an important

experimental paradigm for the future investigations in this field. This work is important for theoretical and experimental cell biologists.

Specific comments of the reviewer are addressed below.

1. The major technological advance here is touted as the first ever analysis of GPCR responses on the single cell level. This is simply not true. Hundreds of studies examined single cell responses of individual cells to GPCR stimulation using by far more precise and interpretable method of patch clamp electrophysiology. This includes classical work on light stimulation of vertebrate and invertebrate photoreceptors, recording of K⁺ channel responses in cardiomyocytes and neurons as well as neurotransmitter modulation of the voltage sensitive Ca²⁺ channels among many others. All of these classical studies used endogenous physiological systems that are by far more defined than Ca²⁺ responses in HEK cells. Furthermore, mathematical modeling of the data had been performed and widely available.

Indeed, single cell analysis has been performed by experimental cell biologists since decades. Yet, before the first attempts to apply the information theory to signal transduction, it has not been performed in the way permitting estimations of the channel capacity of the signaling. For such estimations, i) many single cells must be individually probed, and ii) these same single cells must be probed many times, with different concentrations of the extracellular stimulus. The pioneering papers on the application of the information theory to intracellular signaling, cited and discussed in our work, fulfilled the first requirement, but not the second. We are the first to fulfil both conditions, designing the proper experimental paradigm. This led us to the conclusions provided in our paper.

2. It is widely known and again validated across multiple systems, including work mentioned that GPCRs exhibit gradual responses at single cell level in a dose-dependent fashion, so theoretical motivation and alternatives presented in Figure 1 are entirely misleading. GPCR cascades do not function as binary switches.

The high-profile papers, which laid the ground for the application of the information theory to intracellular signaling, state exactly the opposite: that GPCR cascades (as other signaling systems) function exactly as binary switches. Our work, combining mathematical analysis with experimentation, proves what experimentalists had felt intuitively – that these signaling systems are indeed able, in contrast to the prior analysis, to function as the information transmitting channels, capable of transmitting significantly more information than just a “yes-or-no” response.

3. Based on textbook knowledge the signaling capacity of GPCRs systems should be far beyond 2. The operational model of GPCR introduces the bias factor in addition to potency and efficacy signaling (see reviews by Kenakin) which already make the system pluridimensional. The authors appear to ignore efficacy dimension by normalizing their data to % max. Furthermore, GPCRs use multiple transducers for signaling, e.g. beta arrestins and G proteins which further increases channel capacity- looking only at one readout ignores this complexity.

It appears that the reviewer argues here not against our conclusions, but against those of the previous papers in the field: that intracellular signaling systems operate as binary switches. Our conclusions, in contrast, are that GPCR signaling is a high-capacity transmitter.

4. In addition to the considerations above, analyzing Ca²⁺ response as a proxy for GPCR signaling is far too downstream for being able to conclude how

(Note that the reviewer's comments end here). The reviewer appears to question the validity of measuring Ca²⁺ concentrations as the readout for the selected GPCR signaling, suggesting that it is too downstream. Other reviewers to this paper suggested that it was too upstream. In the process of revisions, we have added a section to the Discussion on this issue. In brief, the pathway certainly has steps before, and steps after Ca²⁺. This fact, however, does not affect the possibility of measuring Ca²⁺ as a meter to determine the channel capacity of the GPCR signaling – all the way from the ligand-receptor interaction down to Ca²⁺ mobilization. I can also add that prior papers have also used this measure in their attempt to estimate the channel capacity.